# Auditing $f$-Differential Privacy in One Run

**Saeed Mahloujifar** [1]   **Luca Melis** [1]   **Kamalika Chaudhuri** [1]

Empirical auditing has emerged as a means of catching some of the flaws in the implementation of privacy-preserving algorithms. Existing auditing mechanisms, however, are either computationally inefficient – requiring multiple runs of the machine learning algorithms —- or suboptimal in calculating an empirical privacy. In this work, we present a tight and efficient auditing procedure and analysis that can effectively assess the privacy of mechanisms. Our approach is efficient; similar to the recent work of Steinke, Nasr, and Jagielski (2023), our auditing procedure leverages the randomness of examples in the input dataset and requires only a single (training) run of the target mechanism. And it is more accurate; we provide a novel analysis that enables us to achieve tight empirical privacy estimates by using the hypothesized $f$-DP curve of the mechanism, which provides a more accurate measure of privacy than the traditional $\epsilon, \delta$ differential privacy parameters. We use our auditing procure and analysis to obtain empirical privacy, demonstrating that our auditing procedure delivers tighter privacy estimates.

## 1. Introduction

Differentially private machine learning (Chaudhuri et al., 2011; Abadi et al., 2016) has emerged as a principled solution to learning models from private data while still preserving privacy. Differential privacy (Dwork, 2006) is a cryptographically motivated definition, which requires an algorithm to possess certain properties: specifically, a randomized mechanism is differentially private if it guarantees that the participation of any single person in the dataset does not impact the probability of any outcome by much.

Enforcing this guarantee requires the algorithm to be carefully designed and analyzed. The process of designing and analyzing such algorithms is prone to errors and imperfections as has been noted in the literature (Tramer et al., 2022). A result of this is that differentially private mechanisms may not perform as intended, either offering less privacy than expected due to flaws in mathematical analysis or implemen-

tation, or potentially providing stronger privacy guarantees that are not evident through a loose analysis.

Empirical privacy auditing (Ding et al., 2018; Nasr et al., 2023; Jagielski et al., 2020) has emerged as a critical tool to bridge this gap. By experimentally assessing the privacy of mechanisms, empirical auditing allows for the verification of privacy parameters. Specifically, an audit procedure is a randomized algorithm that takes an implementation of a mechanism $M$, runs it in a black-box manner, and attempts to test a privacy hypothesis (such as, a differential privacy parameter). The procedure outputs 0 if there is sufficient evidence that the mechanism does not satisfy the hypothesized guarantees and 1 otherwise. The audit mechanism must possess two essential properties: 1) it must have a *provably* small false-negative rate, ensuring that it would not erroneously reject a truly differentially private mechanism, with high probability; 2) it needs to *empirically* exhibit a "reasonable" false positive rate, meaning that when applied to a non-differentially private mechanism, it would frequently reject the privacy hypothesis. The theoretical proof of the false positive rate is essentially equivalent to privacy accounting (Abadi et al., 2016; Dong et al., 2019; Mironov, 2017), which is generally thought to be impossible in a black-box manner (Zhu et al., 2022).

The prior literature on empirical audits of privacy consists of two lines of work, each with its own set of limitations. The first line of work (Ding et al., 2018; Jagielski et al., 2020; Tramer et al., 2022; Nasr et al., 2023) runs a differentially private algorithm multiple times to determine if the privacy guarantees are violated. This is highly computationally inefficient for most private machine learning use-cases, where running the algorithm involves training a large model.

Recent work (Steinke et al., 2023) remove this limitation by proposing an elegant auditing method that runs a differentially private training algorithm a single time. In particular, they rely on the randomness of training data to obtain bounds on the false negative rates of the audit procedure. A key limitation of the approach in (Steinke et al., 2023) is that their audit procedure is sub-optimal in the sense that there is a relatively large gap between the true privacy parameters of mainstream privacy-preserving algorithms (e.g., Gaussian mechanism) and those reported by their auditing algorithm.

In this work, we propose a novel auditing procedure that is

---

[1]Meta. Correspondence to: Saeed Mahloujifar <saeedm@meta.com>.

*Proceedings of the $42^{nd}$ International Conference on Machine Learning*, Vancouver, Canada. PMLR 267, 2025. Copyright 2025 by the author(s).

computationally efficient and accurate. Our method requires only a single run of the privacy mechanism [1] and leverages the $f$-DP curve (Dong et al., 2019), which allows for a more fine-grained accounting of privacy than the traditional reliance on $\epsilon, \delta$ parameters. By doing so, we provide a tighter empirical assessment of privacy.

We experiment with our approach on both simple Gaussian mechanisms as well as a model trained on real data witth DP-SGD. Our experiments show that our auditing procedure can significantly outperform that of (Steinke et al., 2023) (see Figure 1). This implies that better analysis may enable relatively tight auditing of differentially privacy guarantees in a computationally efficient manner in the context of large model training.

**Technical overview:** We briefly summarize the key technical components of our work and compare it with that of Steinke et al. (2023). Their auditing procedure employed a game similar to a membership inference process: the auditor selects a set of canaries and, for each canary, decides whether to inject it into the training set with independent probability 0.5. Once model training is completed, the auditor performs a membership inference attack to determine whether each canary was included. The number of correct guesses made by the adversary in this setting forms a random variable. The key technical contribution of Steinke et al. was to establish a tail bound on this random variable for mechanisms satisfying ($\epsilon$)-DP. Specifically, they demonstrated that the tail of this random variable is bounded by that of a binomial distribution, **binomial**$(n, p)$, where $n$ is the number of canaries and $p = \frac{e^{\epsilon}}{e^{\epsilon}+1}$. To extend this analysis to approximate DP mechanisms, they further showed that the probability of the adversary's success exceeding this tail bound is at most $O(n \cdot \delta)$.

Steinke et al. highlighted a limitation in their approach in auditing specific mechanisms, such as the Gaussian mechanism. They correctly argue that simplifying the mechanism's behavior to just two parameters, $(\epsilon, \delta)$, results in suboptimal auditing of specific mechanisms. In other words, the effectiveness of membership inference attacks against the Gaussian mechanism differs significantly from predictions based solely on the $(\epsilon, \delta)$ parameters. To overcome this limitation, we propose auditing the entire privacy curve of a mechanism, rather than focusing solely on $(\epsilon, \delta)$. Our solution involves three key technical steps:

1. We derive an upper bound on the adversary's success in correctly guessing a specific canary for mechanisms

satisfying $f$-DP. This bound is an improved version of the result by (Hayes et al., 2023) for bounding training data reconstruction in DP mechanisms. However, this is insufficient, as the adversary's guesses could be dependent, potentially leading to correlated successes (e.g., correctly or incorrectly guessing all samples).

2. To address the issue of dependency, we refine our analysis by defining $p_i$ as the probability of the adversary making exactly $i$ correct guesses. We derive a recursive relation that bounds $p_i$ based on $p_1, \ldots, p_{i-1}$. This recursive bound is the main technical novelty of our work. To derive this bound, we consider two conditions: the adversary correctly guesses the first canary or not. In the first case, we use our analysis from Step 1 to bound the probability of making $i - 1$ correct guesses given that the first guess was correct. For the incorrect guess case, we perform a combinatorial analysis to eliminate the condition. This analysis uses the fact that shuffling of the canaries does not change the probabilities of making $i$ correct guesses. We note that it is crucial not to use the analysis of Step 1 for both cases. This is because the analysis of Step 1 cannot be tight for both cases at the same time. Finally, leveraging the convexity of trade-off functions and applying Jensen's inequality, we derive our final recursive relation. To the best of our knowledge, This combination of trade-off function with shuffling is a new technique and could have broader applications.

3. Finally, we design an algorithm that takes advantage of the recursive relation to numerically calculate an upper bound on the tail of the distribution. The algorithm is designed carefully so that we do not need to invoke the result of step 2 for very small events.

We also generalize our analysis to a broader notion of canary injection and membership inference. Specifically, we utilize a reconstruction game where the auditor can choose among $k$ options for each canary point, introducing greater entropy for each choice. This generalization allows for auditing mechanisms with fewer canaries.

In the rest of the paper, we first introduce the notions of $f$-DP and explain what auditing based on $f$-DP entails. We then present our two auditing procedures based on membership inference and reconstruction attacks (Section 2). In Section 3, we provide a tight analysis of our audit's accuracy based on $f$-DP curves. Finally, in Section 4, we describe the experimental setup used to compare the bounds.

## 2. Auditing $f$- differential privacy

Auditing privacy involves testing a "privacy hypothesis" about an algorithm $M$. Different mathematical forms can be

---

[1]In the context of privacy-preserving training of machine learning models, the privacy mechanism refers to the training algorithm. Therefore, when we mention a single run, we are specifically referring to a single execution of the training algorithm, not the inference algorithm.

used for a "privacy hypothesis," but they all share the common characteristic of being about an algorithm/mechanism $M$. For example, one possible hypothesis is that applying SGD with specific hyperparameters satisfies some notion of privacy. With this in mind, the privacy hypothesis are often mathematical constraints on the sensitivity of the algorithm's output to small changes in its input. The most well-known definition among these is differential privacy.

**Definition 2.1.** A mechanism $M$ is $(\epsilon, \delta)$-DP if for all neighboring datasets $\mathcal{S}, \mathcal{S}'$ with $|\mathcal{S}\Delta\mathcal{S}'| = 1$ and all measurable sets $T$, we have $\Pr[M(\mathcal{S}) \in T] \leq e^\epsilon \Pr[M(\mathcal{S}') \in T] + \delta$.

In essence, differential privacy ensures that the output distribution of the algorithm does not heavily depend on a single data point. Based on this definition, one can hypothesize that a particular algorithm satisfies differential privacy with certain $\epsilon$ and $\delta$ parameters. Consequently, auditing differential privacy involves designing a test for this hypothesis. We will later explore the desired properties of such an auditing procedure. However, at present, we recall a stronger notion of privacy known as $f$-differential privacy.

**Notation** For a function $f\colon [0,1] \to [0,1]$ we use $\bar{f}$ to denote the function $\bar{f}(x) = 1 - f(x)$.

**Definition 2.2.** A mechanism $\mathcal{M}$ is $f$-DP if for all neighboring datasets $\mathcal{S}, \mathcal{S}'$ and all $|\mathcal{S}\Delta\mathcal{S}'| = 1$ measurable sets $T$ we have

$$\Pr[M(\mathcal{S}) \in T] \leq \bar{f}\big(\Pr[M(\mathcal{S}')] \in T]\big).$$

Note that this definition generalizes the notion of approximate differential privacy by allowing a more complex relation between the probability distributions of $M(S)$ and $M(S')$. The following proposition shows how one can express approximate DP as an instantiation of $f$-DP.

**Proposition 2.3.** *A mechanism is $(\epsilon, \delta)$-DP if it is $f$-DP with respect to $\bar{f}(x) = e^\epsilon \cdot x + \delta$.*

Although the function $f$ could be an arbitrary function, without loss of generality, we only consider a specific class of functions in this notion.

*Remark* 2.4. Whenever we say that a mechanism satisfies $f$-DP, we implicitly imply that $f$ is a valid trade-off function . That is, $f$ is defined on domain $[0,1]$ and has a range of $[0,1]$. Moreover, $f$ is a decreasing and convex with $f(x) \leq 1 - x$ for all $x \in [0,1]$. We emphasize that this is without loss of generality. That is, if a mechanism is $f$-DP for a an arbitrary function $f : [0,1] \to [0,1]$, then it is also $f'$-DP for valid trade-off function $f'$ with $f'(x) \leq f(x)$ for all $x \in [0,1]$ (See Proposition 2.2 in (Dong et al., 2019)).

**Definition 2.5** (Order of $f$-DP curves). For two trade-off functions $f_1$ and $f_2$, we say $f_1$ is more private than $f_2$ and denote it by $f_1 \geq f_2$ iff $f_1(x) \geq f_2(x)$ for all $x \in$ $[0,1]$. Also, for a family of trade-off functions $F$, we use $maximal(F)$ to denote the set of maximal elements w.r.t to the privacy relation. Note that $F$ could be a partial ordered set, and $maximal(F)$ may have multiple elements.

Now that we have defined our privacy hypothesis, we can turn our attention to auditing these notions.

**Definition 2.6** (Auditing $f$-DP). An audit procedure takes the description of a mechanism $\mathcal{M}$, a trade-off function $f$, and outputs a bit that determines whether the mechanism satisfies $f$-DP or not. We define it as a two-step procedure.

- $game\colon M \to O$, In this step, the auditor runs a potentially randomized experiment/game using the description of mechanism $\mathcal{M} \in M$ and obtains some observation $o \in O$.

- $evaluate : O \times F \to \{0,1\}$, In this step, the auditor will output a bit $b$ based on an observation $o$ and a trade-off function $f$. This audit operation tries to infer whether the observation $o$ is "likely" for a mechanism that satisfies $f$-DP.

The audit procedure is $\psi$-accurate if for all mechanism $\mathcal{M}$ that satisfy $f$-DP, we have

$$\Pr_{o \leftarrow game(\mathcal{M})}[evaluate(o, f) = 1] \geq \psi.$$

Note that we are defining the accuracy only for positive cases. This is the only guarantee we can get from attacks. For guarantees in negative cases, we need to perform privacy accounting for the mechanism (Wang et al., 2023).

Next, we formally define the notion of empirical privacy (Nasr et al., 2021) based on an auditing procedure. This notion provides the best privacy guarantee that is not violated by auditors' observation from a game setup.

**Definition 2.7** (Empirical Privacy). Let $(game, evaluate)$ be an audit procedure. We define the empirical privacy random variable for a mechanism $\mathcal{M}$, w.r.t a family $F$ of trade-off functions, to be the output of the following process. We first run the game to obtain observation $o = game(\mathcal{M})$. We then construct

$$F_o = maximal(\{f \in F; evaluate(o, f) = 1\})$$

where the maximal set is constructed according to Definition 2.5. Then, the empirical privacy of the mechanism at a particular $\delta$ is defined as

$$\epsilon(\delta) = \min_{f \in F_o} \max_{x \in [0,1]} log(\frac{1 - f(x) - \delta}{x}).$$

Note that the empirical privacy $\epsilon(\delta)$ is a function of the observation $o$. Since, $o$ itself is a random variable, then $\epsilon(\delta)$ is also a random variable.

**How to choose the family of trade-off functions?** The family of trade-off functions should be chosen based on the expectations of the true privacy curve. For example, if one expects the privacy curve of a mechanism to be similar to that of a Gaussian mechanism, then they would choose the set of all trade-off functions imposed by a Gaussian mechanism as the family. For example, many believe that in the hidden state model of privacy (Ye & Shokri, 2022), the final model would behave like a Gaussian mechanism with higher noise than what is expected from the accounting in the white-box model (where we assume we release all the intermediate models). Although we may not be able to prove this hypothesis , we can use our framework to calculate the empirical privacy, while assuming that the behavior of the final model would be similar to that of a Gaussian mechanism.

**Auditing $f$-DP vs DP:** $f$-DP can be viewed as a collection of DP parameters, where instead of considering $(\epsilon, \delta)$ as fixed scalars, we treat $\epsilon$ as a function of $\delta$. For any $\delta \in [0, 1]$, there exists an $\epsilon(\delta)$ such that the mechanism satisfies $(\epsilon(\delta), \delta)$-DP. The $f$-DP curve effectively represents the entire privacy curve rather than a single $(\epsilon, \delta)$ pair. Thus, auditing $f$-DP can be expected to be more effective, as there are more constraints that need to be satisfied. A naive approach for auditing $f$-DP is to perform an audit for approximate DP at each $(\epsilon, \delta)$ value along the privacy curve, rejecting if any of the audits fail. However, this leads to sub-optimal auditing performance. First, the auditing analysis involves several inequalities that bound the probabilities of various events using differential privacy guarantees. The probability of these events could take any number between $[0, 1]$. Using a single $(\epsilon, \delta)$ value to bound the probability of all these events cannot be tight because the linear approximation of privacy curve is tight in at most a single point. Hence, the guarantees of $(\epsilon, \delta)$-DP cannot be simultaneously tight for all events. However, with $f$-DP, we can obtain tight bounds on the probabilities of all events simultaneously. Second, For each $(\epsilon, \delta)$ we have a small possibility of incorrectly rejecting the privacy hypothesis. So if we audit privacy for $(\epsilon(\delta), \delta)$ independently, we will reject any privacy hypothesis with probability 1.0. This challenge can be potentially resolved by using correlated randomness.

To demonstrate this key difference, we try a baseline for d auditing $f$-DP based on the work of (Steinke et al., 2024b)[2]. In this baseline, we consider a gaussian mechanism with noise $\sigma$. Then, we audit the privacy curve at various values of $\delta$. For this, we need to make sure that we run the attack once (the correlated randomness mentioned above), so we fix the number of guesses to be the optimal choice for $\delta = 10^{-5}$. Then we observe the attack's performance and apply

---

[2]This experiment was suggested by our anonymous reviewer. We than the reviewer for their suggestion.

the method of (Steinke et al., 2024b). We observe that this improves the performance over the plain method but there it still has large gap with direct $f$-DP auditing. The details and results of this experiment are reported in Section 4.2.

## 2.1. Guessing games

Here, we introduce the notion of guessing games which is a generalization of membership inference attacks (Nasr et al., 2023), and closely resembles the reconstruction setting introduced in (Hayes et al., 2023).

**Definition 2.8.** Consider a mechanism $M : [k]^m \to \Theta$. In a guessing game we first sample an input dataset $\mathbf{u} \in [k]^m$ from an arbitrary distribution. We run the mechanism to get $\theta \sim M(\mathbf{u})$. Then a guessing adversary $A : \Theta \to ([k] \cup \{\bot\})^m$ tries to guess the input to the mechanism from the output. We define

- the number of guesses by $c' = \sum_{i=1}^{m} \mathbf{I}(A(\theta)_i \neq \bot)$

- and the number of correct guesses by $c = \sum_{i=1}^{m} \mathbf{I}(A(\theta)_i = \mathbf{u}_i)$.

Then we output $(c, c')$ as the output of the game.

These guessing games are integral to our auditing strategies. We outline two specific ways to instantiate the guessing game. The first procedure is identical to that described in the work of (Steinke et al., 2023) and resembles membership inference attacks. The second auditing algorithm is based on the reconstruction approach introduced by (Hayes et al., 2023). In Section 3, we present all of our results in the context of the general notion of guessing games, ensuring that our findings extend to both the membership inference and reconstruction settings.

**Auditing by membership inference:** Algorithm 1 describes a game setup based on membership inference attacks. In this setup, we have a fixed training set $\mathcal{T}$ and a set of canaries $\mathcal{C}$. We first sample a subset $\mathcal{S}$ of the canaries using poisson sampling. Then we run the mechanism $\mathcal{M}$ on $\mathcal{T} \cup \mathcal{S}$ to get a model $\theta \sim \mathcal{M}(\mathcal{T} \cup \mathcal{S})$. Then the adversary $A$ inspects $\theta$ and tries to find examples that were present in $\mathcal{S}$. Observe that this procedure is a guessing game with $k = 2$ and $m = |\mathcal{C}|$. This is simply because the adversary is guessing between two choices for each canary, it is either included or not included. Note that this procedure is modular, we can use any $\mathcal{T}$ and $\mathcal{C}$ for the training set and canary set. We can also use any attack algorithm $A$.

We note that membership inference attacks have received a lot of attention recently (Homer et al., 2008; Shokri et al., 2017; Leino & Fredrikson, 2020; Bertran et al., 2024; Hu et al., 2022; Matthew et al., 2023; Duan et al., 2024; Zarifzadeh et al., 2023). These attack had a key difference

from our attack setup and that is the fact that there is only a single example that the adversary is trying to make the inference for. Starting from the work of (Shokri et al., 2017), researchers have tried to improve attacks in various settings (Ye et al., 2022; Zarifzadeh et al., 2023). For example, using calibration techniques has been an effective way to improve membership inference attacks (Watson et al., 2021; Carlini et al., 2022). Researchers have also changed their focus from average case performance of the attack to the tails of the distribution and measured the precision at low recall values (Ye et al., 2022; Nasr et al., 2021).

A substantial body of research has also explored the relationship between membership inference attacks and differential privacy (Sablayrolles et al., 2019; Mahloujifar et al., 2022; Balle et al., 2022; Bhowmick et al., 2018; Stock et al., 2022; Balle et al., 2022; Guo et al., 2022; Kaissis et al., 2023; 2024), using this connection to audit differential privacy (Steinke et al., 2024a; Pillutla et al., 2024; Jagielski et al., 2020; Ding et al., 2018; Bichsel et al., 2018; Nasr et al., 2021; 2023; Steinke et al., 2024b; Tramer et al., 2022; Bichsel et al., 2021; Lu et al., 2022; Andrew et al., 2023; Cebere et al., 2024; Annamalai & De Cristofaro, 2024; Chadha et al., 2024). Some studies have investigated empirical methods to prevent membership inference attacks that do not rely on differential privacy (Hyland & Tople, 2019; Jia et al., 2019; Chen & Pattabiraman, 2023; Li et al., 2024; Tang et al., 2022; Nasr et al., 2018). An intriguing avenue for future research is to use the concept of empirical privacy to compare the performance of these empirical methods with provable methods, such as DP-SGD.

---

**Algorithm 1** Membership inference in one run game

**input** Oracle access to a mechanism $\mathcal{M}(\cdot)$, A training dataset $\mathcal{T}$, An indexed canary set $\mathcal{C} = \{x_i; i \in [m]\}$, An attack algorithm $A$.

---

1: Set $m = |\mathcal{C}|$
2: Sample $u = (u_1, \ldots, u_m) \sim \text{Bernoulli}(0.5)^m$, a binary vector where $u_i = 1$ with probability 0.5.
3: Let $\mathcal{S} = \{\mathcal{C}[u_i]; u_i = 1\}_{i \in [m]}$, the subset of selected elements in $\mathcal{C}$.
4: Run mechanism $M$ on $\mathcal{T} \cup \mathcal{S}$ to get output $\theta$.
5: Run membership inference attack $A$ on $\theta$ to get set of membership predictions $v = (v_1, \ldots, v_m)$ which is supported on $\{0, 1, \perp\}^m$.
6: Count $c$, the number of correct guesses where $u_i = v_i$ and $c'$ the total number of guesses where $v_i \neq \perp$.
   **return** $(c, c')$.

---

**Auditing by reconstruction:** We also propose an alternative way to perform auditing by reconstruction attacks. This setup starts with a training set $\mathcal{S}_t$, similar to the membership inference setting. Then, we have a family of $m$

canary sets $\{\mathcal{S}_c^i; i \in [m]\}$ where each $\mathcal{S}_c^i$ contains $k$ distinct examples. Before training, we construct a set $\mathcal{S}_s$ of size $m$ by uniformly sampling an example from each $\mathcal{S}_c^i$. Then, the adversary tries to find out which examples were sampled from each canary set $\mathcal{S}_c^i$ by inspecting the model. We recognize that this might be different from what one may consider a true "reconstruction attack", because the adversary is only performing a selection. However, if you consider the set size to be arbitrary large, and the distribution on the set to be arbitrary, then this will be general enough to cover various notions of reconstruction. We also note that (Hayes et al., 2023) use the same setup to measure the performance of the reconstruction attacks.

---

**Algorithm 2** Reconstruction in one run game

**input** Oracle access to a mechanism $\mathcal{M}(\cdot)$, A training dataset $\mathcal{T}$, number of canaries $m$, number of options for each canary $k$, a matrix of canaries $\mathcal{C} = \{x_j^i\}_{i \in [m], j \in [k]}$, an attack algorithm $A$.

---

1: Let $u = (u_1, \ldots, u_m)$ be a vector uniformly sampled from $[k]^m$.
2: Let $\mathcal{S} = \{x_{u_i}^i\}_{i \in [m]}$.
3: Run mechanism $\mathcal{M}$ on $\mathcal{S} \cup \mathcal{T}$ to get output $\theta$.
4: Run a reconstruction attack $A$ on $\theta$ to get a vector $v = (v_1, \ldots, v_m)$ which is a vector in $([k] \cup \{\perp\})^m$.
5: Count $c$ the number of coordinates where $u_i = v_i$ and $c'$ the number of coordinates where $v_i \neq \perp$.
   **return** $(c, c')$.

---

# 3. Implications of $f$-DP for guessing games

In this section, we explore the implications of $f$-DP for guessing games. Specifically, we focus on bounding the probability of making more than $c$ correct guesses for adversaries that make at most $c'$ guesses. We begin by stating our main theorem, followed by an explanation of how it can be applied to audit the privacy of a mechanism.

**Theorem 3.1.** *[Bounds for adversary with bounded guesses] Let $M : [k]^m \to \Theta$ be a $f$-DP mechanism. Let $\mathbf{u}$ be a random variable uniformly distributed on $[k]^m$. Let $A : \Theta \to ([k] \cup \{\perp\})^m$ be a guessing adversary which always makes at most $c'$ guesses, that is*

$$\forall \theta \in \Theta, \Pr\left[\left(\sum_{i=1}^{m} I(A(\theta)_i \neq \perp)\right) > c'\right] = 0,$$

*and let $\mathbf{v} \equiv A(M(\mathbf{u}))$. Define $p_i = \Pr\left[\left(\sum_{j \in [m]} \mathbf{I}(\mathbf{u}_j = \mathbf{v}_j)\right) = i\right]$. For all subset of indices $T \subseteq [c']$, we have*

$$\sum_{i \in T} \frac{i}{m} p_i \leq \bar{f}\left(\frac{1}{k-1} \sum_{i \in T} \frac{c' - i + 1}{m} p_{i-1}\right).$$

This Theorem, which we consider to be our main technical contribution, provides a nice invariant that bounds the probability $p_i$ (probability of making exactly $i$ correct guesses) based on the value of other $p_j$s. Imagine $P_f$ to be a set of vectors $p = (p_1, \ldots, p_{c'})$ that could be realized for an attack on a $f$-DP mechanism. Theorem 3.1 significantly confines this set. However, this still does not resolve the auditing task. We are interested in bounding $\max_{p \in P_f} \sum_{i=c}^{c'} p_i$, the maximum probability that an adversary can make more than $c$ correct guesses for an $f$-DP mechanism. Next, we show how we can algorithmically leverage the limitations imposed by Theorem 3.1 and calculate an upper bound on $\max_{p \in P_f} \sum_{i=c}^{c'} p_i$.

### 3.1. Numerically bounding the tail

In this subsection, we specify our procedure for bounding the tail of the distribution and hence the accuracy of our auditing procedure. Our algorithm needs oracle access to $f$ and $\bar{f}$ and decides an upper bound on the probability of an adversary making $c$ correct guesses in a guessing game with alphabet size $k$ and a mechanism that satisfies $f$-DP. This algorithm relies on the confinement imposed by Theorem 3.1. Note that Algorithm 3 is a decision algorithm, it takes a value $\tau$ and decide if the probability of making more than $c$ correct guesses is less than or equal to $\tau$. We can turn this algorithm to a estimation algorithm by performing a binary search on the value of $\tau$. However, for our use cases, we are interested in a fixed $\tau$. This is because we (similar to (Steinke et al., 2023)) want to set the accuracy of our audit to be a fixed value such as 0.95.

---

**Algorithm 3** Numerically deciding an upper bound probability of making more than $c$ correct guesses

---

**input** Oracle access to $\bar{f}$ and $\bar{f}^{-1}$, number of guesses $c'$, number of correct guesses $c$, number of samples $m$, alphabet size $k$, probability threshold $\tau$ (default is $\tau = 0.05$).

1: $\forall 0 \leq i \leq c$ set $h[i] = 0$, and $r[i] = 0$.
2: set $r[c] = \tau \cdot \frac{c}{m}$.
3: set $h[c] = \tau \cdot \frac{c'-c}{m}$.
4: **for** $i \in [c-1, \ldots, 0]$ **do**
5: $\quad h[i] = (k-1)\bar{f}^{-1}\big(r[i+1]\big)$
6: $\quad r[i] = r[i+1] + \frac{i}{c'-i} \cdot \big(h[i] - h[i+1]\big)$.
7: **end for**
8: **if** $r[0] + h[0] \geq \frac{c'}{m}$ **then**
9: $\quad$ Return True; (Probability of $c$ correct guesses (out of $c'$) is less than $\tau$).
10: **else**
11: $\quad$ Return False; (Probability of having $c$ correct guesses (out of $c'$) could be more than $\tau$).
12: **end if**

---

**Theorem 3.2.** *If Algorithm 3 returns True on inputs*

$\bar{f}, k, m, c, c'$ *and* $\tau$, *then for any* $f$-DP *mechanism* $M: [k]^m \to \Theta$, *any guessing adversary* $A: \Theta \to ([k] \cup \{\bot\})^m$ *with at most* $c'$ *guesses, defining* $\mathbf{u}$ *to be uniform over* $[k]^m$, *and setting* $\mathbf{v} \equiv A\big(M(\mathbf{u})\big)$, *we have* $\Pr[\big(\sum_{i=1}^m \mathbf{I}(\mathbf{u}_i = \mathbf{v}_i)\big) \geq c] \leq \tau$.

In a nutshell, this algorithm tries to obtain an upper bound on the sum $p_c + p_{c+1} + \ldots, p_{c'}$. We assume this probability is greater than $\tau$, and we obtain lower bound on $p_{c-1} + p_c + \cdots + p_{c'}$ based on this assumption. We keep doing this recursively until we have a lower bound on $p_0 + \cdots + p_{c'}$. If this lower bound is greater than 1, then we have a contradiction and we return true. The detailed proof of this Theorem is involved and requires careful analysis. We defer the full proof of Theorem to appendix.

**Auditing $f$-DP with Algorithm 3:** When auditing the $f$-DP for a mechanism, we assume we have injected $m$ canaries, and ran an adversary that is allowed to make $c'$ guesses and recorded that the adversary have made $c$ correct guesses. In such scenario, we will reject the hypothesized privacy of the mechanism if the probability of this observation is less than a threshold $\tau$, which we by default set to 0.05. To this end, we just call Algorithm 3 with parameters $c$, $c'$, $m$, $\tau = 0.05$ and $f$. Then if the algorithm returns *True*, we will reject the privacy hypothesis and approve it otherwise.

**Empirical privacy:** Although auditing in essence is a hypothesis testing, previous work has used auditing algorithms to calculate empirical privacy as defined in definition 2.7. In this work, we follow the same route. For simplicity, we only consider an ordered set of privacy hypotheses $h_1, \ldots, h_w$ as our family of $f$-DP curves. These sets are ordered in their strength, meaning that any mechanism that satisfies $h_i$, would also satisfy $h_j$ for all $j < i$. Then, we would report the strongest privacy hypothesis that passes the test as the empirical privacy of the mechanism.

## 4. Experiments

Most of our experiments are conducted in an *idealized setting*, similar to that used in (Steinke et al., 2023), unless otherwise stated. In this setting, the attack success rate is automatically calculated to simulate the expected number of correct guesses by an optimal adversary (details of the idealized setting are provided in Algorithm 4 in Appendix). We then use this expected number as the default value for the number of correct guesses to derive the empirical $\epsilon$. More specifically, as specified in Definition 2.6, we instantiate our auditing with a game and evaluation setup. We use Algorithm 4 in Appendix as our game setup. This algorithm returns the number of guesses and the number of correct guesses as the observations from the game. Then, we use Algorithm 3 as our evaluation setup to audit an $f$-DP curve based on the observation from Algorithm 4. Note that in our

comparison with the auditing of Steinke et al., we always use the same membership inference game setup ($k = 2$) as defined in their work. This ensures that our comparison is only on the evaluation part of the audit procedure.

In all experiments, we use empirical $\epsilon$ as the primary metric for evaluating our bounds.

**$f$-DP candidates:** As described in Section 3.1 , we need an ordered set of $f$-DP curves to obtain empirical privacy. In our experiments, we use $f$-DP curves for Gaussian mechanisms with varying standard deviations (this forms an ordered set because the $f$-DP curve of a Gaussian mechanism with a higher standard deviation dominates that of a lower standard deviation). For sub-sampled Gaussian mechanisms, the ordered set consists of $f$-DP curves for sub-sampled Gaussian mechanisms with fixed sub-sampling rates and number of steps, and various noise stds.

### 4.1. Comparison with (Steinke et al., 2023)

In this section, we evaluate our auditing method for membership inference in an idealized setting, using the work of (Steinke et al., 2023) as our main baseline. We compare our approach directly to their work, which operates in the same setting as ours.

**Simple Gaussian Mechanism:** In the first experiment (Figure 1), we audit a simple Gaussian mechanism, varying the standard deviations from $[0.5, 1.0, 2.0, 4.0]$, resulting in different theoretical $\epsilon$ values. We vary the number of canaries ($m$) from $10^2$ to $10^7$ for auditing, set the bucket size to $k = 2$, and adjust the number of guesses ($c'$) for each number of canaries. For each combination of $m, c'$, and each standard deviation, we calculate ($c$) using Algorithm 4 (the idealized setting in appendix). This algorithm calculates the expected number of correct guesses for an adversary who observes the output of an $m$-dimensional gaussian mechanism, $V + \mathcal{N}(0^m, \sigma)$, with $V$ being a uniform sample from $\{0, 1\}^m$. The adversary's goal is to guess $c'$ coordinates in $V$. $c$ is calculated to be the expected number of correct guesses by the optimal adversary. Note that this setup is designed as the worst-case scenario for the gaussian mechanism. After obtaining $c$, we then audit all tuples of $(m, c, c')$ using the $f$-DP curves of the Gaussian mechanism. Then we find the $c$ that achieves the highest empirical $\epsilon$ and then report that as the empirical $\epsilon$. We audit the exact same setup with the auditing method of (Steinke et al., 2024b). Figure 1 demonstrates that our approach outperforms the empirical privacy results from Steinke et al. Interestingly, while the bound in Steinke et al. (2023) degrades as the number of canaries increases, our bounds continue to improve.

**Experiments on CIFAR-10:** We also run experiments on CIFAR-10 on a modified version of the WRN16-

4 (Zagoruyko & Komodakis, 2016) architecture, which substitutes batch normalization with group normalization. We follow the setting proposed by (Sander et al., 2023), which use custom augmentation multiplicity (i.e., random crop around the center with 20 pixels padding with reflect, random horizontal flip and jitter) and apply an exponential moving average of the model weights with a decay parameter of 0.9999. We run white-box membership inference attacks by following the strongest attack used in the work of (Steinke et al., 2023), where the auditor injects multiple canaries in the training set with crafted gradients. More precisely, each canary gradient is set to zero except at a single random index ("Dirac canary" (Nasr et al., 2023)). Note that in the white-box attack, the auditor has access to all intermediate iterations of DP-SGD. The attack scores are computed as the dot product between the gradient update during consecutive model iterates and the aggregated gradients from dp-sgd. As done in the work of (Steinke et al., 2023), we audit CIFAR-10 model with $m = 5,000$ canaries and all training points from CIFAR-10 $n = 50,000$ for the attack. We set the batch size to $4,096$, using augumented multiplicity of $K = 16$ and training for $2,500$ DP-SGD steps. For $\varepsilon = 8.0, \delta = 10^{-5}$, we achieved 77% accuracy when auditing, compared to 80% without injected canaries. Figure 2 shows the comparison between the auditing scheme by (Steinke et al., 2023) with ours for different values of theoretical $\varepsilon$. We are able to achieve tighter empirical lower bounds. We also report the performance of the black-box attack, where the auditor does not control the training pipeline and can only compute membership scores (losses) from the final model. Figure 3 shows how we achieve tighter lower bounds compared to Steinke et al. (2023) where we set $m = 1,000$ and all training samples are used for auditing ($m = n$). This corresponds to the stronger setup for the black-box auditor in Steinke et al. (2023).

Finally, we report the results of auditing the robust membership inference attack (Zarifzadeh et al., 2023) (RMIA), which to the best of our knowledge, represents the State-of-The-Art (SoTA) black-box membership inference attack on CIFAR-10 from the literature. We reproduce the results in (Zarifzadeh et al., 2023) with a non-private WideResNet model (with depth 28 and width 2) for 100 training epochs on half of the dataset chosen at random resulting on a test accuracy of 92.2%. We run the low-cost black-box membership inference attack using 2 reference models in the offline setting (Zarifzadeh et al., 2023). We audit with $m = 5,000$ canaries and report in Figure 4 the comparison between our scheme and (Steinke et al., 2023) with different abstention values. Our auditing method clearly outperforms Steinke et al. for all bounded guesses settings, with higher empirical epsilon for larger abstention values (i.e., fewer guesses).

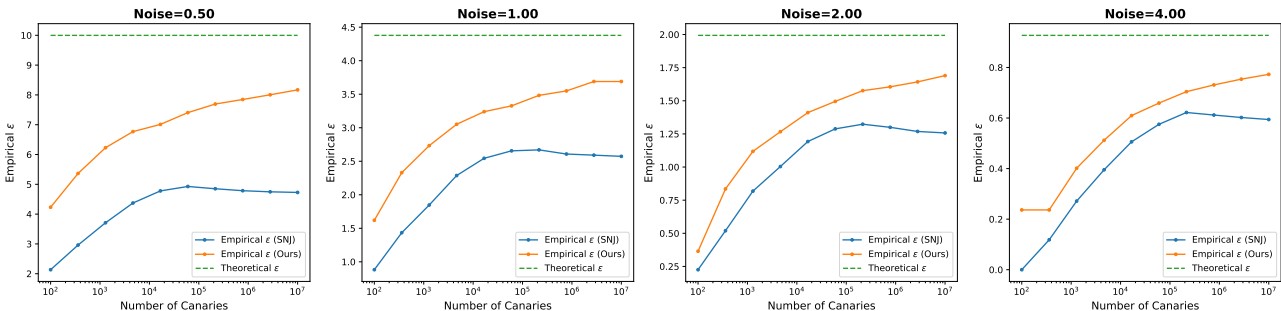

*Figure 1.* Comparison between our empirical privacy lower bounds and that of (Steinke et al., 2023) at $\delta = 10^{-5}$.

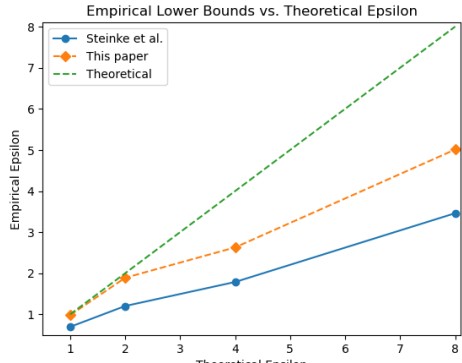

*Figure 2.* Comparison with auditing procedure of (Steinke et al., 2023) on CIFAR-10 in white-box setting using gradient-based membership inference. Empirical $\epsilon$ is reported at $\delta = 10^{-5}$.

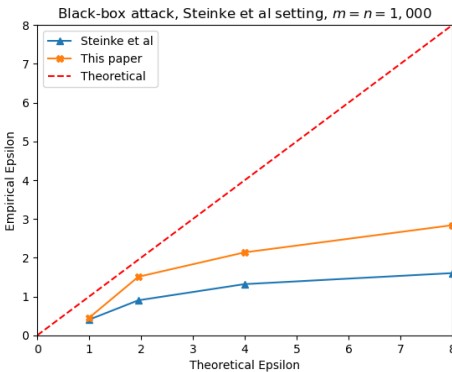

*Figure 3.* Comparison with Steinke et al. (2023) for CIFAR-10 in black-box setting. Empirical $\epsilon$ is reported at $\delta = 10^{-5}$.

**Why is our bound better better than (Steinke et al., 2023)?** The bounds in Steinke et al. audit approximate DP. That is, they take DP parameters $(\epsilon, \delta)$ and prove an upper bound on the probability of any adversary obtaining $c'$ correct guesses out of $c$ total guesses, given $m$ canaries

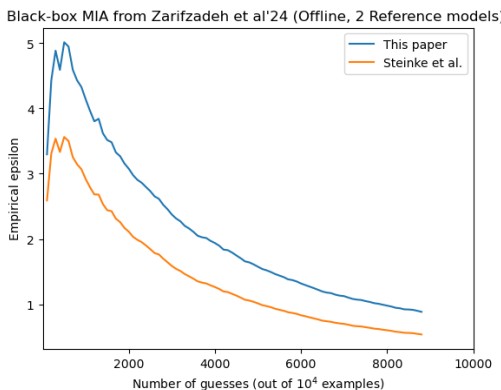

*Figure 4.* Comparison with auditing procedure of (Steinke et al., 2023) on non-private model trained on CIFAR-10 against black-box RMIA method (Zarifzadeh et al., 2023). Empirical $\epsilon$ is reported at $\delta = 10^{-5}$.

available. For the case of $\delta = 0$, their bound is tight. For the case of $\delta > 0$, however, they need to define a set of undesirable events and bound their collective probability. This incurs an additional $O(m \cdot \delta)$ in the probability. The reason why their bounds start to degrade when we increase $m$ is this very fact. The $m \cdot \delta$ term starts to dominate and causes the empirical epsilon estimation to become worse. The reason we do not observe this behavior is that we do not use $(\epsilon, \delta)$ to approximate the privacy curve, we use the exact curve as is. As we know, the linear approximation of privacy curve is optimal only in a single point for mechanisms that we are interested in (e.g. the Gaussian mechanism). Namely, there is only a single probability $p' \in [0, 1]$ where we have

$$p = \Pr[M(D) \in E] \quad \text{and} \quad e^\epsilon \cdot p + \delta = \Pr[M(D') \in E].$$

Our bound is designed to avoid this issue. We derive a bound that uses the exact $f$-DP curve, which ensures that for all probabilities $p \in [0, 1]$ the upper bound on the blow-up of events of size $p$ is tight. Moreover, the way we invoke our Theorem 3.1 in our numerical estimation 3 is designed to

| Noise | # Canaries | Theoretical | Steinke et al. | Steinke et al. (pointwise) | Ours |
|---|---|---|---|---|---|
| $\sigma = 0.5$ | $10^5$ | 9.99 | 4.99 | 5.01 | 8.16 |
| $\sigma = 1.0$ | $10^5$ | 4.37 | 2.61 | 2.71 | 3.61 |
| $\sigma = 2.0$ | $10^5$ | 1.99 | 1.33 | 1.35 | 1.59 |
| $\sigma = 4.0$ | $10^6$ | 0.92 | 0.61 | 0.67 | 0.82 |

*Table 1.* Comparison of empirical privacy Gaussian noise levels. The reported numbers of are empirical $\epsilon$ at $\delta = 10^{-5}$.

apply the bound on events that can be simultaneously tight. This way, our bound does not have the problem of getting worse as the number of samples increases.

Note that this does not mean that there is no way to improve our bound. We still see some gap between the empirical epsilon and the true epsilon. The reason for this, we believe, is in the way numerical tail bound in Algorithm 3.2 is designed. In this algorithm, we make some relaxations that can be a source of sub-optimality. Specifically, our analysis benefits from the fact that the expectation of correct guesses, conditioned on the correct guesses being greater than $c$ divided by the expectation incorrect guesses conditioned on the same event is greater than $c/c'$. This step is not tight as we cannot have a mechanism where the adversary makes exactly $c$ correct guesses with probability greater than 0, while making more than $c$ correct guesses with probability exactly 0. For a more interested reader, Equations 6 and 7 in the proof of Theorem 3.2 is a source of sub-optimality that future work can resolve.

### 4.2. Improving (Steinke et al., 2024b) by testing multiple hypothesis.

In this section, we describe a method that uses the method of (Steinke et al., 2024b) to audit $f$-DP. We use the idea that if a mechanism satisfies $f$-DP, then for all $\delta \in [0, 1]$ it should pass the DP audit for $(\epsilon_\delta, \delta)$, where $\epsilon_\delta$ is the optimal $\epsilon$ obtained from $f$ for $\delta$. A key issue here is that auditing in one run will always suffer from probabilistic error. There is a small chance $\tau$ that the audit mechanism rejects the privacy hypothesis incorrectly. When doing the test multiple times, then we have to multiply the the failure probability by the number of trials.

However, we can avoid this by using shared randomness between trials. Specifically, if we only run the privacy game once and use the output of the game to audit privacy for different values of $(\epsilon, \delta)$, we can potentially avoid this multiplication. Here, we design an experiment that shows even with this this approach, the bounds of previous work cannot match ours. We try to auditing Gaussian DP. First we instantiate a membership inference game with a fixed number of canaries $(m)$ and a fixed number of guesses $(c')$. This is optimized to achieve the best $\epsilon$ at $\delta = 10^{-5}$. We collect the number of correct guesses $(c)$ in the membership inference

game. Using $(m, c, c')$ we can now auditing $(\epsilon_\delta, \delta)$-DP for a large range of values of $\delta$ ($\delta = 10^{-x}$ for 60 different values of $x$ linearly spread between 3 to 9), where $\epsilon_\delta$ is the privacy of a gaussian mechanism with a given noise at $\delta$. Then, we reject the privacy hypothesis for gaussian-DP if any of the individual tests are rejected. Using this auditing procedure, we obtain empirical epsilon values.

Table 1 shows the results of our experiments. We can see that there is still a large gap between our auditing and the multiple run of the approach of previous work as described above. As discussed in Section 2, the reason for the multiple testing method being inferior to our direct $f$-DP auditing is that in the multiple DP-auditing approach, each auditing procedure is oblivious to other points on the $f$-DP curve and can only observe a single point on the curve. Whereas for our method, the audit procedure observes the entire curve. This point has also been discussed by the authors of (Steinke et al., 2024b) as a limitation on of their approach.

### 5. Conclusions and limitations

We introduce a new approach for auditing the privacy of algorithms in a single run using $f$-DP curves. This method enables more accurate approximations of the true privacy guarantees, addressing the risk of a "false sense of privacy" that may arise from previous approximation techniques. By leveraging the entire $f$-DP curve, rather than relying solely on point estimates, our approach provides a more nuanced understanding of privacy trade-offs. This allows practitioners to make more informed decisions regarding privacy-utility trade-offs in real-world applications. However, our approach does not provide a strict upper bound on privacy guarantees but instead offers an estimate of the privacy parameters that can be expected in practical scenarios. We also recognize that, despite the improvements over prior work, we still observe a gap between the empirical and theoretical privacy reported in the "one run" setting. Future work could focus on closing this gap to further enhance the reliability of empirical privacy estimations.

## Impact Statement

This paper aims to advance the empirical measurement of algorithmic privacy. By improving our ability to evaluate the privacy risks associated with machine learning and data processing systems, this work contributes to the development of more trustworthy and accountable AI technologies. The main societal benefit is positive: practitioners and policymakers will be better equipped to assess and mitigate potential privacy harms, leading to safer deployment of data-driven systems.

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

# A. Proofs

## A.1. Proof outline for Theorem 3.1

In this subsection, we outline the main ingredients we need to prove our Theorem 3.1. We also provide a warm up proof for a simplified version of Theorem 3.1 without abstentions and then we focus on the proof of the main theorem. First, we have a Lemma that bounds the probability of any event conditioned on correctly guessing a single canary.

**Lemma A.1.** *Let $M : [k]^m \to \Theta$ be a mechanism that satisfies $f$-DP. Also let $A\colon \Theta \to ([k] \cup \{\bot\})^m$ be a guessing attack. Let $\mathbf{u}$ be a random variable uniformly distributed over $[k]^m$ and let $\mathbf{v} \equiv A\big(M(\mathbf{u})\big)$. Then for any subset $E \subseteq \Theta$ we have*

$$f_k'' \Big( \Pr\big[M(\mathbf{u}) \in E\big] \Big) \leq \Pr\big[M(\mathbf{u}) \in E \text{ and } u_1 = v_1\big] \leq f_k' \Big( \Pr\big[M(\mathbf{u}) \in E\big] \Big)$$

*where*

$$f_k'(x) = \sup\{\alpha; \alpha + f(\frac{x - \alpha}{k - 1}) \leq 1\} \ \ and \ \ f_k''(x) = \inf\{\alpha; (k - 1)f(\alpha) + x - \alpha) \leq 1\}.$$

This Lemma which is a generalization and an improvement over the main Theorem of (Hayes et al., 2023), shows that the probability of an event cannot change too much if we condition on the success of adversary on one of the canaries. Note that this Lemma immediately implies a bound on the expected number of correct guesses by any guessing adversary (by just using linearity of expectation). However, here we are not interested in expectations. Rather, we need to derive tail bounds. The proof of Theorem 3.1 relies on some key properties of the $f'$ and $f''$ functions defined in the statement of Lemma A.1. These properties are specified in the following Proposition and proved in the Appendix.

**Proposition A.2.** *The functions $f_k'$ as defined in Lemma A.1 is increasing and concave. The function $f_k''$ as defined in Lemma A.1 is increasing and convex.*

Now, we are ready to outline the proof of a simplified variant of our Theorem 3.1 for adversaries that make a guess on all canaries. This makes the proof much simpler and enables us to focus more on the key steps in the proof.

**Theorem A.3** (Special case of 3.1). *Let $M : [k]^m \to \Theta$ be a $f$-DP mechanism. Let $\mathbf{u}$ be a random variable uniformly distributed on $[k]^m$. Let $A\colon \Theta \to [k]^m$ be a guessing adversary and let $\mathbf{v} \equiv A(M(\mathbf{u}))$. Define $p_i = \Pr\Big[(\sum_{j\in[m]} \mathbf{I}\big(\mathbf{u}_j = \mathbf{v}_j\big)) = i\Big]$. For all subset of indices $T \subseteq [m]$, we have*

$$\sum_{i\in T} \frac{i}{m} p_i \leq \bar{f}(\frac{1}{k-1} \sum_{i\in T} \frac{m-i+1}{m} p_{i-1})$$

*Proof.* Let us define a random variable $\mathbf{t} = (\mathbf{t}_1, \ldots, \mathbf{t}_m)$ which is defined as $\mathbf{t}_i = \mathbf{I}(\mathbf{u}_i = \mathbf{v_i})$ We have

$$p_c = \Pr[\sum_{i=1}^m \mathbf{t}_i = c] = \Pr[\sum_{i=2}^m \mathbf{t}_i = c - 1 \text{ and } \mathbf{t}_1 = 1] + \Pr[\sum_{i=2}^m \mathbf{t}_i = c \text{ and } \mathbf{t}_1 = 0]$$

Now by Lemma A.1 we have $\Pr[\sum_{i=2}^m \mathbf{t}_i = c - 1 \text{ and } \mathbf{t}_1 = 1] \leq f_k'(\sum_{i=2}^m \mathbf{t}_i = c - 1)$. This is a nice invariant that we can use but $\sum_{i=2}^m \mathbf{t}_i = c - 1$ could be really small depending on how large $m$ is. To strengthen the bound we sum all $p_c$'s for $c \in T$, and then apply the lemma on the aggregate. That is

$$\sum_{j\in T} p_j = \sum_{j\in T} \Pr[\sum_{i=1}^m \mathbf{t}_i = j] = \sum_{j\in T} \Pr[\sum_{i=2}^m \mathbf{t}_i = j \text{ and } \mathbf{t}_1 = 0] + \sum_{j\in T} \Pr[\sum_{i=2}^m \mathbf{t}_i = j - 1 \text{ and } \mathbf{t}_1 = 1]$$

$$= \Pr[\sum_{i=2}^m \mathbf{t}_i \in T \text{ and } \mathbf{t}_1 = 0] + \Pr[1 + \sum_{i=2}^m \mathbf{t}_i \in T \text{ and } \mathbf{t}_1 = 1]$$

Now we only use the inequality from Lemma A.1 for the second quantity above. Using the inequality for both probabilities is not ideal because they cannot be tight at the same time. So we have,

$$\sum_{j\in T} p_j \leq \Pr[\sum_{i=2}^m \in T \text{ and } \mathbf{t}_1 = 0] + f_k'(\Pr[1 + \sum_{i=2}^m \mathbf{t}_i \in T]).$$

Now we use a trick to make this cleaner. We use the fact that this inequality is invariant to the order of indices. So we can permute $\mathbf{t_i}$'s and the inequality still holds. We have,

$$\sum_{j \in T} p_j \leq \mathop{\mathrm{E}}_{\pi \sim \Pi[m]}[\Pr[\sum_{i=2}^{m} \mathbf{t}_{\pi(i)} \in T \text{ and } \mathbf{t}_{\pi(1)} = 0]] + \mathop{\mathrm{E}}_{\pi \sim \Pi[m]}[f_k'(\Pr[1 + \sum_{i=2}^{m} \mathbf{t}_{\pi(i)} \in T])]$$

$$\leq \mathop{\mathrm{E}}_{\pi \sim \Pi[m]}[\Pr[\sum_{i=2}^{m} \mathbf{t}_{\pi(i)} \in T \text{ and } \mathbf{t}_{\pi(1)} = 0]] + f_k'(\mathop{\mathrm{E}}_{\pi \sim \Pi[m]}[\Pr[1 + \sum_{i=2}^{m} \mathbf{t}_{\pi(i)} \in T]]).$$

Now we perform a double counting argument. Note that when we permute the order $\sum_{i=2}^{m} \mathbf{t}_{\pi(i)} = j$ and $\mathbf{t}_{\pi(1)} = 0$ counts each instance $t_1, \ldots, t_m$ with exactly $j$ non-zero locations, for exactly $(m - j) \times (m - 1)!$ times. Therefore, we have

$$\mathop{\mathrm{E}}_{\pi \sim \Pi[m]}[\Pr[\sum_{i=2}^{m} \mathbf{t}_{\pi(i)} \in T \text{ and } \mathbf{t}_{\pi(1)} = 0]] = \sum_{j \in T} \frac{m - j}{m} p_j.$$

With a similar argument we have,

$$\mathop{\mathrm{E}}_{\pi \sim \Pi[m]}[\Pr[1 + \sum_{i=2}^{m} \mathbf{t}_{\pi(i)} \in T]] = \sum_{j \in T} \frac{m - j + 1}{m} p_{j-1} + \frac{j}{m} p_j.$$

Then, we have

$$\sum_{j \in T} p_j \leq \sum_{j \in T} \frac{m - j}{m} p_j + f_k'(\sum_{j \in T} \frac{j}{m} p_j + \frac{m - j + 1}{m} p_{j-1}).$$

And this implies

$$\sum_{j \in T} \frac{j}{m} p_j \leq f_k'(\sum_{j \in T} \frac{j}{m} p_j + \frac{m - j + 1}{m} p_{j-1}).$$

And this, by definition of $f_k'$ implies

$$\sum_{j \in T} \frac{j}{m} p_j \leq \bar{f}(\frac{1}{k - 1} \sum_{j \in T} \frac{m - j + 1}{m} p_{j-1}).$$

$\square$

## A.2. Proof of Main Lemmas and Theorems

*Proof of Lemma A.1.* Let $p = \Pr[M(\mathbf{u}) \in E \text{ and } u_1 = v_1]$ and $q = \Pr[M(\mathbf{u}) \in E]$. We have

$$p = \sum_{i \in [k]} \Pr[M(\mathbf{u}) \in E \text{ and } u_1 = v_1 = i]$$

$$= \frac{1}{k} \sum_{i \in [k]} \Pr[M(\mathbf{u}) \in E \text{ and } v_1 = i \mid u_1 = i]$$

$$= \frac{1}{k} \sum_{i \in [k]} \frac{1}{k-1} \Big( \sum_{j \in [k] \backslash \{i\}} \Pr[M(\mathbf{u}) \in E \text{ and } v_1 = i \mid u_1 = i] \Big)$$

$$\text{(By definition of } f\text{-DP)} \quad \leq \frac{1}{k} \sum_{i \in [k]} \frac{1}{k-1} \Big( \sum_{j \in [k] \backslash \{i\}} 1 - f\big( \Pr[M(\mathbf{u}) \in E \text{ and } v_1 = i \mid u_1 = j] \big) \Big)$$

$$\text{(By convexity of } f\text{)} \quad \leq 1 - f \left( \frac{1}{k} \sum_{i \in [k]} \frac{1}{k-1} \Big( \sum_{j \in [k] \backslash \{i\}} \Pr[M(\mathbf{u}) \in E \text{ and } v_1 = i \mid u_1 = j] \Big) \right)$$

$$= 1 - f \left( \frac{1}{k-1} \sum_{i \in [k]} \Big( \sum_{j \in [k] \backslash \{i\}} \frac{1}{k} \Pr[M(\mathbf{u}) \in E \text{ and } v_1 = i \mid u_1 = j] \Big) \right)$$

$$= 1 - f \left( \frac{1}{k-1} \sum_{i \in [k]} \Big( \sum_{j \in [k] \backslash \{i\}} \Pr[M(\mathbf{u}) \in E \text{ and } v_1 = i \text{ and } u_1 = j] \Big) \right)$$

$$= 1 - f \big( \frac{1}{k-1} \Pr[M(\mathbf{u}) \in E \text{ and } u_1 \neq v_1] \big)$$

$$= 1 - f \big( \frac{q-p}{k-1} \big).$$

Similarly we have,

$$p = \sum_{i \in [k]} \Pr[M(\mathbf{u}) \in E \text{ and } u_1 = v_1 = i]$$

$$= \frac{1}{k} \sum_{i \in [k]} \Pr[M(\mathbf{u}) \in E \text{ and } v_1 = i \mid u_1 = i]$$

$$= \frac{1}{k} \sum_{i \in [k]} \frac{1}{k-1} \Big( \sum_{j \in [k] \backslash \{i\}} \Pr[M(\mathbf{u}) \in E \text{ and } v_1 = i \mid u_1 = i] \Big)$$

$$\text{(By definition of } f\text{-DP)} \quad \geq \frac{1}{k} \sum_{i \in [k]} \frac{1}{k-1} \Big( \sum_{j \in [k] \backslash \{i\}} f^{-1}\big( 1 - \Pr[M(\mathbf{u}) \in E \text{ and } v_1 = i \mid u_1 = j] \big) \Big)$$

$$\text{(By convexity of } f\text{)} \quad \geq f^{-1} \left( \frac{1}{k} \sum_{i \in [k]} \frac{1}{k-1} \Big( \sum_{j \in [k] \backslash \{i\}} 1 - \Pr[M(\mathbf{u}) \in E \text{ and } v_1 = i \mid u_1 = j] \Big) \right)$$

$$= f^{-1} \left( \frac{1}{k-1} \sum_{i \in [k]} \Big( \sum_{j \in [k] \backslash \{i\}} \frac{1}{k}(1 - \Pr[M(\mathbf{u}) \in E \text{ and } v_1 = i \mid u_1 = j]) \Big) \right)$$

$$= f^{-1} \left( \frac{1}{k-1} \sum_{i \in [k]} \Big( \sum_{j \in [k] \backslash \{i\}} \Pr[M(\mathbf{u}) \in E \text{ and } v_1 = i \text{ and } u_1 = j] \Big) \right)$$

$$= f^{-1} \big( \frac{1}{k-1}(1 - \Pr[M(\mathbf{u}) \in E \text{ and } u_1 \neq v_1]) \big)$$

$$= f^{-1} \big( \frac{1-q+p}{k-1} \big).$$

This implies that,

$$f(p) \cdot (k-1) + q - p \le 1$$

$\square$

*Proof of Proposition A.2.* The function is increasing simply because $f$ is decreasing. We now prove concavity. Let $\alpha_1 = f_k(x_1)$ and $\alpha_2 = f_k(x_2)$. By definition of $f_k$ we have

$$\alpha_1 + f(\frac{x_1 - \alpha_1}{k-1}) \le 1$$

and

$$\alpha_2 + f(\frac{x_2 - \alpha_2}{k-1}) \le 1.$$

Averaging these two we get,

$$\frac{\alpha_1 + \alpha_2}{2} + \frac{f(\frac{x_1 - \alpha_1}{k-1}) + f(\frac{x_2 - \alpha_2}{k-1})}{2} \le 1$$

By convexity of $f$ we have

$$\frac{\alpha_1 + \alpha_2}{2} + f(\frac{\frac{x_1 + x_2}{2} - \frac{\alpha_1 + \alpha_2}{2}}{k-1}) \le 1$$

Therefore, by definition of $f_k'$, we have $f_k'(\frac{x_1 + x_2}{2}) \ge \frac{\alpha_1 + \alpha_2}{2}$. Similarly, $f_k''$ in increasing just because $f$ is decreasing. And assuming $\alpha_1 = f_k(x_1)$ and $\alpha_2 = f_k(x_2)$ we have

$$f_k''(\frac{x_1 + x_2}{2}) \le \frac{\alpha_1 + \alpha_2}{2}$$

which implies $f_k''$ is convex. $\square$

*Proof of Theorem 3.1.* Instead of working with an adversary with $c'$ guesses, we assume we have an adversary that makes a guess on all $m$ inputs, however, it also submits a vector $\mathbf{q} \in \{0,1\}^m$, with exactly $c'$ 1s and $m - c'$ 0s. So the output of this adversary is a vector $\mathbf{v} \in [k]^m$ and a vector $\mathbf{q} \in \{0,1\}^m$. Then, only correct guesses that are in locations that $\mathbf{q}$ is non-zero is counted. That is, if we define a random variable $\mathbf{t} = (\mathbf{t}_1, \ldots, \mathbf{t}_m)$ as $\mathbf{t}_i = \mathbf{I}(\mathbf{u}_i = \mathbf{v_i})$ then we have

$$p_c = \Pr[\sum_{i=1}^m \mathbf{t}_i \cdot \mathbf{q}_i = c]$$

$$= \Pr[\sum_{i=2}^m \mathbf{t}_i = c - 1 \text{ and } \mathbf{t}_1 = 1 \text{ and } \mathbf{q}_1 = 1] + \Pr[\sum_{i=2}^m \mathbf{t}_i = c \text{ and } \mathbf{t}_1 \cdot \mathbf{q}_1 = 0]$$

Now by Lemma A.1 we have

$$\Pr[\sum_{i=2}^m \mathbf{t}_i = c - 1 \text{ and } \mathbf{t}_1 = 1 \text{ and } \mathbf{q}_1 = 1] \le f_k'(\sum_{i=2}^m \mathbf{t}_i = c - 1 \text{ and } \mathbf{q}_1 = 1).$$

This is a nice invariant that we can use but $\sum_{i=2}^m \mathbf{t}_i = c - 1$ could be really small depending on how large $m$ is. To strengthen the bound we sum all $p_c$'s for $c \in T$, and then apply the lemma on the aggregate. That is

$$\sum_{j \in T} p_j = \sum_{j \in T} \Pr[\sum_{i=1}^m \mathbf{t}_i = j]$$

$$= \sum_{j \in T} \Pr[\sum_{i=2}^m \mathbf{t}_i = j \text{ and } \mathbf{t}_1 \cdot \mathbf{q}_1 = 0] + \sum_{j \in T} \Pr[\sum_{i=2}^m \mathbf{t}_i = j - 1 \text{ and } \mathbf{t}_1 = 1 \text{ and } \mathbf{q}_1 = 1]$$

$$= \Pr[\sum_{i=2}^m \mathbf{t}_i \in T \text{ and } \mathbf{t}_1 \cdot \mathbf{q}_1 = 0] + \Pr[1 + \sum_{i=2}^m \mathbf{t}_i \in T \text{ and } \mathbf{t}_1 = 1 \text{ and } \mathbf{q}_1 = 1]$$

Now we only use the inequality from Lemma A.1 for the second quantity above. Using the inequality for both probabilities is not ideal because they cannot be tight at the same time. So we have,

$$\sum_{j \in T} p_j \le \Pr[\sum_{i=2}^{m} \in T \text{ and } \mathbf{t}_1 \cdot \mathbf{q}_1 = 0] + f'_k(\Pr[1 + \sum_{i=2}^{m} \mathbf{t}_i \in T \text{ and } \mathbf{q}_1 = 1]).$$

Now we use a trick to make this cleaner. We use the fact that this inequality is invariant to the order of indices. So we can permute $\mathbf{t_i}$'s and the inequality still holds. We have,

$$\sum_{j \in T} p_j \le \operatorname*{E}_{\pi \sim \Pi[m]}[\Pr[\sum_{i=2}^{m} \mathbf{t}_{\pi(i)} \in T \text{ and } \mathbf{t}_{\pi(1)} \cdot \mathbf{q}_{\pi(1)} = 0]] + \operatorname*{E}_{\pi \sim \Pi[m]}[f'_k(\Pr[1 + \sum_{i=2}^{m} \mathbf{t}_{\pi(i)} \in T])]$$

$$\le \operatorname*{E}_{\pi \sim \Pi[m]}[\Pr[\sum_{i=2}^{m} \mathbf{t}_{\pi(i)} \in T \text{ and } \mathbf{t}_{\pi(1)} = 0]] + f'_k(\operatorname*{E}_{\pi \sim \Pi[m]}[\Pr[1 + \sum_{i=2}^{m} \mathbf{t}_{\pi(i)} \in T \text{ and } \mathbf{q}_{\pi(1)} = 1]]).$$

Now we perform a double counting argument. Note that when we permute the order $\sum_{i=2}^{m} \mathbf{t}_{\pi(i)} = j$ and $\mathbf{t}_{\pi(1)} = 0$ counts each instance $t_1, \dots, t_m$ with exactly $j$ non-zero locations, for exactly $(m - j) \times (m - 1)!$ times. Therefore, we have

$$\operatorname*{E}_{\pi \sim \Pi[m]}[\Pr[\sum_{i=2}^{m} \mathbf{t}_{\pi(i)} \cdot \mathbf{q}_{\pi(i)} \in T \text{ and } \mathbf{t}_{\pi(1)} \cdot \mathbf{q}_{\pi(i)} = 0]] = \sum_{j \in T} \frac{m - j}{m} p_j.$$

With a similar argument we have,

$$\operatorname*{E}_{\pi \sim \Pi[m]}[\Pr[1 + \sum_{i=2}^{m} \mathbf{t}_{\pi(i)} \cdot \mathbf{q}_{\pi(i)} \in T \text{ and } \mathbf{q}_{\pi(1)} = 1]] = \sum_{j \in T} \frac{c' - j + 1}{m} p_{j-1} + \frac{j}{m} p_j.$$

Then, we have

$$\sum_{j \in T} p_j \le \sum_{j \in T} \frac{m - j}{m} p_j + f'_k(\sum_{j \in T} \frac{j}{m} p_j + \frac{c' - j + 1}{m} p_{j-1})$$

$$= \sum_{j \in T} \frac{m - j}{m} p_j + f'_k(\sum_{j \in T} \frac{j}{m} p_j + \frac{c' - j + 1}{m} p_{j-1}).$$

And this implies

$$\sum_{j \in T} \frac{j}{m} p_j \le f'_k(\sum_{j \in T} \frac{j}{m} p_j + \frac{c' - j + 1}{m} p_{j-1}).$$

And this, by definition of $f'_k$ implies

$$\sum_{j \in T} \frac{j}{m} p_j \le \bar{f}(\frac{1}{k - 1} \sum_{j \in T} \frac{c' - j + 1}{m} p_{j-1}).$$

$\square$

*Proof of Theorem 3.2.* To prove Theorem 3.2, we first state and prove a lemma which is consequence of Theorem 3.1.

**Lemma A.4.** *For all $c \le c' \in [m]$ let us define*

$$\alpha_c = \sum_{i=c}^{c'} \frac{i}{m} p_i \quad and \quad \beta_c = \sum_{i=c}^{c'} \frac{c' - i}{m} p_i$$

*We also define a family of functions $r = \{r_{i,j} : [0, 1] \times [0, 1] \to [0, 1]\}_{i \le j \in [m]}$ and $h = \{h_{i,j} : [0, 1] \to [0, 1]\}$ that are defined recursively as follows.*

$\forall i \in [m] : r_{i,i}(\alpha, \beta) = \alpha$ and $h_{i,i}(\alpha, \beta) = \beta$ and for all $i < j$ we have

$$h_{i,j}(\alpha, \beta) = (k-1)\bar{f}^{-1}\Big(r_{i+1,j}(\alpha, \beta)\Big)$$

$$r_{i,j}(\alpha, \beta) = r_{i+1,j}(\alpha, \beta) + \frac{i}{c'-i}(h_{i,j}(\alpha, \beta) - h_{i+1,j}(\alpha, \beta))$$

*Then for all $i \leq j$ we have*

$$\alpha_i \geq r_{i,j}(\alpha_j, \beta_j) \quad and \quad \beta_i \geq h_{i,j}(\alpha_j, \beta_j)$$

*Moreover, for $i < j$, $r_{i,j}$ and $h_{i,j}$ are increasing with respect to their first argument and decreasing with respect to their second argument.*

*Proof of Lemma A.4.* We prove this by induction on $j - i$. For $j - i = 0$, the statement is trivially correct. We have

$$h_{i,j}(\alpha_j, \beta_j) = (k-1)\bar{f}^{-1}(r_{i+1,j}(\alpha_j, \beta_j)).$$

By induction hypothesis, we have $r_{i+1,j}(\alpha_j, \beta_j) \leq \alpha_{i+1}$. Therefore we have

$$h_{i,j}(\alpha_j, \beta_j) \leq (k-1)\bar{f}^{-1}(\alpha_{i+1}). \tag{1}$$

Now by invoking Theorem 3.1, we have

$$\alpha_{i+1} \leq \bar{f}(\frac{\beta_i}{k-1}).$$

Now since $\bar{f}$ is increasing, this implies

$$(k-1)\bar{f}^{-1}(\alpha_{i+1}) \leq \beta_i \tag{2}$$

Now putting, inequalities 1 and 2 together we have $h_{i,j}(\alpha_j, \beta_j) \leq \beta_i$. This proves the first part of the induction hypothesis for the function $h$. Also note that $h_{i,j}$ is increasing in its first component and decreasing in the second component by invoking induction hypothesis and the fact that $\bar{f}^{-1}$ is increasing. Now we focus on function $r_{i,j}$. First note that there is an alternative form for $r_{i,j}$ by opening up the recursive relation. Let $\gamma_z = \frac{z}{c'-z} - \frac{z-1}{c'-z+1}$. We have ,

$$r_{i,j}(\alpha, \beta) = r_{j,j}(\alpha, \beta) + \frac{i}{c'-i}h_{i,j}(\alpha, \beta) - \frac{j-1}{c'-j+1}h_{j,j}(\alpha, \beta) + \sum_{z=i+1}^{j-1} \gamma_z h_{z,j}(\alpha, \beta)$$

$$= r_{j,j}(\alpha, \beta) + \frac{i}{c'-i}h_{i,j}(\alpha, \beta) - \frac{j}{c'-j}h_{j,j}(\alpha, \beta) + \sum_{z=i+1}^{j} \gamma_z h_{z,j}(\alpha, \beta)$$

$$= \alpha - \frac{j}{c'-j}\beta + \frac{i}{c'-i}h_{i,j}(\alpha, \beta) + \sum_{z=i+1}^{j} \gamma_z h_{z,j}(\alpha, \beta). \tag{3}$$

Now we show that for all $i$ we have

$$\alpha_i = \frac{i}{c'-i}\beta_i + \sum_{z=i+1}^{m} \gamma_z \beta_z. \tag{4}$$

This is because we have

$$\alpha_i - \frac{i}{c'-i}\beta_i = \sum_{z=i+1}^{c'} (\frac{z}{m} - \frac{i(c'-z)}{(c'-i)m})p_z.$$

On the other hand we have

$$\sum_{z=i+1}^{m} \gamma_z \beta_z = \sum_{z=i+1}^{m} \left( \sum_{z'=i+1}^{z} \gamma_{z'} \right) \frac{c'-z}{m} p_z$$

$$= \sum_{z=i+1}^{m} \left( \frac{z}{c'-z} - \frac{i}{c'-i} \right) \frac{c'-z}{m} p_z$$

$$= \sum_{z=i+1}^{m} \left( \frac{z}{m} - \frac{i(c'-z)}{(c'-i)m} \right) p_z$$

and this shows that Equation 4 is correct. Therefore for all $i < j$ we have

$$\alpha_i - \alpha_j = \frac{i}{c'-i} \beta_i - \frac{j}{c'-j} \beta_j + \sum_{z=i+1}^{j} \gamma_z \beta_z$$

Now, using the induction hypothesis for $h$ we have,

$$\alpha_i \geq \alpha_j + \frac{i}{c'-i} h_{i,j}(\alpha_j, \beta_j) - \frac{j}{c'-j} \beta_j + \sum_{z=i+1}^{j} \gamma_z h_{z,j}(\alpha_j, \beta_j). \tag{5}$$

Now verify that the right hand side of Equation 5 is equal to $r_{i,j}(\alpha_j, \beta_j)$ by the formulation of Equation 3

Also, using the induction hypothesis, we can observe that the right hand side of 3 is increasing in $\alpha_j$ and decreasing in $\beta_j$ because all terms there are increasing in $\alpha_j$ and decreasing in $\beta_j$. $\qquad \square$

This lemma enables us to prove that algorithm 3 is deciding a valid upper bound on the probability correctly guessing $c$ examples out of $c'$ guesses. To prove this, assume that the probability of such event is equal to $\tau'$, Note that this means $\alpha_c + \beta_c = \frac{c'}{m}\tau'$. Also note that

$$\frac{\alpha_c}{\beta_c} \geq \frac{c}{c'-c} \tag{6}$$

therefore, we have

$$\alpha_c \geq \frac{c}{m}\tau' \tag{7}$$

and $\beta_c \leq \frac{c'-c}{m}\tau'$. Therefore, using Lemma A.1 we have $\alpha_0 \geq r_{0,c}(\frac{c}{m}\tau', \frac{c'-c}{m}\tau')$ and $\beta_0 \geq h_{0,c}(\frac{c}{m}\tau', \frac{c'-c}{m}\tau')$.

Now we prove a lemma about the function $s_{i,j}(\tau) = h_{i,j}(\frac{c}{m}\tau, \frac{c'-c}{m}\tau) + r_{i,j}(\frac{c}{m}\tau, \frac{c'-c}{m}\tau)$.

**Lemma A.5.** *the function $s_{i,j}(\tau) = h_{i,j}(\frac{c}{m}\tau, \frac{c'-c}{m}\tau) + r_{i,j}(\frac{c}{m}\tau, \frac{c'-c}{m}\tau)$ is increasing in $\tau$ for $i < j \leq c$.*

*Proof.* To prove this, we show that for all $i < j \leq c$ both $r_{i,j}(\frac{c}{m}\tau, \frac{c'-c}{m}\tau)$ and $h_{i,j}(\frac{c}{m}\tau, \frac{c'-c}{m}\tau)$ are increasing in $\tau$. We prove this by induction on $j - i$. For $j - i = 1$, we have

$$h_{i,i+1}\left(\frac{c}{m}\tau, \frac{c'-c}{m}\tau\right) = (k-1)\bar{f}^{-1}\left(\frac{c}{m}\tau\right).$$

We know that $\bar{f}^{-1}$ is increasing, therefore $h_{i,i+1}(\frac{c}{m}\tau, \frac{c'-c}{m}\tau)$ is increasing in $\tau$ as well. For $r_{i,i+1}$ we have

$$r_{i,i+1}\left(\frac{c}{m}\tau, \frac{c'-c}{m}\tau\right) = \frac{c}{m}\tau + \frac{i}{c'-i}\left(h_{i,i+1}\left(\frac{c}{m}\tau, \frac{c'-c}{m}\tau\right) - \frac{c'-c}{m}\tau\right)$$

So we have

$$r_{i,i+1}(\frac{c}{m}\tau, \frac{c'-c}{m}\tau) = \frac{c(c'-i)-i(c'-c)}{m(c'-i)}\tau + \frac{i}{c'-i}h_{i,i+1}(\frac{c}{m}\tau, \frac{c'-c}{m}\tau)$$
$$= \frac{(c-i)c'}{m(c'-i)}\tau + \frac{i}{c'-i}h_{i,i+1}(\frac{c}{m}\tau, \frac{c'-c}{m}\tau).$$

We already proved that $h_{i,i+1}(\frac{c}{m}\tau, \frac{c'-c}{m}\tau)$ is increasing in $\tau$. We also have $\frac{(c-i)c'}{m(c'-i)} > 0$, since $i < c$. Therefore

$$r_{i,i+1}(\frac{c}{m}\tau, \frac{c'-c}{m}\tau)$$

is increasing in $\tau$. So the base of induction is proved. Now we focus on $j - i > 1$. For $h_{i,j}$ we have

$$h_{i,j}(\frac{c}{m}\tau, \frac{c'-c}{m}\tau) = (k-1)\bar{f}^{-1}(r_{i+1,j}(\frac{c}{m}\tau, \frac{c'-c}{m}\tau)).$$

By the induction hypothesis, we know that $r_{i+1,j}(\frac{c}{m}\tau, \frac{c'-c}{m}\tau)$ is increasing in $\tau$, and we know that $\bar{f}^{-1}$ is increasing, therefore, $h_{i,j}(\frac{c}{m}\tau, \frac{c'-c}{m}\tau)$ is increasing in $\tau$.

For $r_{i,j}$, note that we rewrite it as follows

$$r_{i,j}(\alpha, \beta) = \alpha - \frac{j}{c'-j}\beta + \sum_{z=i}^{j-1}\lambda_z \cdot h_{z,j}(\alpha, \beta)$$

where $\lambda_z = (\frac{z+1}{c'-z-1} - \frac{z}{c'-z}) \geq 0$. Therefore, we have

$$r_{i,j}(\frac{c}{m}\tau, \frac{c'-c}{m}\tau) = \tau(\frac{c}{m} - \frac{(c'-c)j}{m(c'-j)}) + \sum_{z=i}^{j-1}\lambda_z \cdot h_{z,j}(\frac{c}{m}\tau, \frac{c'-c}{m}\tau)$$
$$= \tau\frac{c'(c-j)}{m(c'-j)} + \sum_{z=i}^{j-1}\lambda_z \cdot h_{z,j}(\frac{c}{m}\tau, \frac{c'-c}{m}\tau).$$

Now we can verify that all terms in this equation are increasing in $\tau$, following the induction hypothesis and the fact that $\lambda_z > 0$ and also $j \leq c$. □

Now using this Lemma, we finish the proof. Note that we have $\alpha_0 + \beta_0 = \frac{c'}{m}$.

So assuming that $\tau' \geq \tau$, then we have

$$\frac{c'}{m} = \alpha_0 + \beta_0 \geq s_{0,c}(\tau') \geq s_{0,c}(\tau).$$

The last step of algorithm checks if $s_{0,c} \geq \frac{c'}{m}$ and it concludes that $\tau' \leq \tau$ if that's the case, because $s_{0,c}$ is increasing in $\tau$. This means that the probability of having more than $c$ guesses cannot be more than $\tau$. □

## B. Ablation Experiments

**Reconstruction attacks:** To show the effect of the bucket size ($k$) on the auditing performance, in Figure 5, we change the number of examples in the two different setups. In first setup we use 10,000 canaries and change the bucket size from 50 to 5000. In the other setup we only use 100 canaries and change the bucket-size from 3 to 50. Note that in these experiments, we do not use abstention and only consider adversaries that guess all examples.

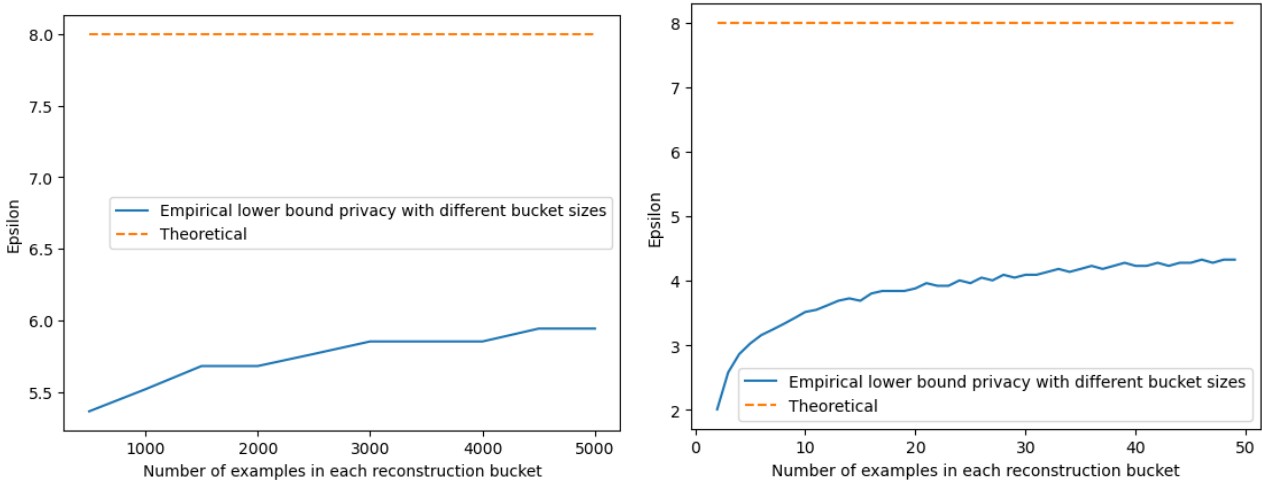

*Figure 5.* Effect of bucket size on the empirical lower bounds for reconstruction attack (Gaussian mechanism with standard deviation 0.6). Left: 10,000 canaries with bucket size up-to 5000. Right: 100 canaries with bucket-size up-to 50. Empirical $\epsilon$ is reported at $\delta = 10^{-5}$.

**Effect of number of guesses** In Figures 6–9, we compare the theoretical upper bound, our lower bound, and the lower bound of Steinke et al. with varying number of guesses. In total, we have $m = 10^7$ canaries. The number of correct guesses is determined using Algorithm 4 (the idealized setting). Then we use our and (Steinke et al., 2023)'s auditing with the resulting numbers and report the empirical $\epsilon$. As we can see, both our and Steinke et al.'s auditing procedure achieve the best auditing performance for small number of guesses. This shows the importance of abstention in auditing.

A curious reader might wonder why the number of guesses has such a big impact on empirical privacy. Essentially, our analysis involves estimating how many correct guesses an adversary can make when given a certain number of attempts. We focus on specific percentiles of these distributions. The accuracy of our empirical privacy estimates can vary significantly based on how much the number of correct guesses fluctuates, which is influenced by how many guesses we allow the adversary to make. To explain further, consider a random variable representing the ratio of correct guesses ($c$) to total guesses ($c'$). If we reduce the number of guesses, the variance of this ratio tends to decrease because the ratio approaches 1 (the adversary can make more correct guesses when we decrease $c'$). Conversely, if we increase the number of guesses, the variance can also decrease because having more guesses generally leads to a more stable average, owing to the law of large numbers. This balance makes the number of guesses a crucial factor in optimizing for the best estimate of empirical privacy.

**Varying $\delta$ and confidence levels:** We also examine the effect of $\delta$ on the obtained empirical $\epsilon$. We fix the number of canaries to $10^5$ and the number of guesses to $1,500$ and the number of correct guesses are set to $1,429$, suggested by the idealized setting. We use a Gaussian mechanism with standard deviation $1.0$, we vary the value of $\delta$ and the confidence level to observe how they affect the results. Figures 10 and 11 shows the bound of (Steinke et al., 2023) and our bound, respectively. Note that our lower bounds represent the true behavior of $\delta$ independent of the confidence level, in contrast to the bound of (Steinke et al., 2023).

## C. Other datasets

We also report in Figure 12 our privacy analysis method in the black-box attack setting on the tabular dataset of shopping records Purchase (Shokri et al., 2017). We replicate the same setup in (Zarifzadeh et al., 2023), on a non-private MLP model trained on 25000 samples for 50 epochs. We outperform Steinke et al. method for all numbers of guesses

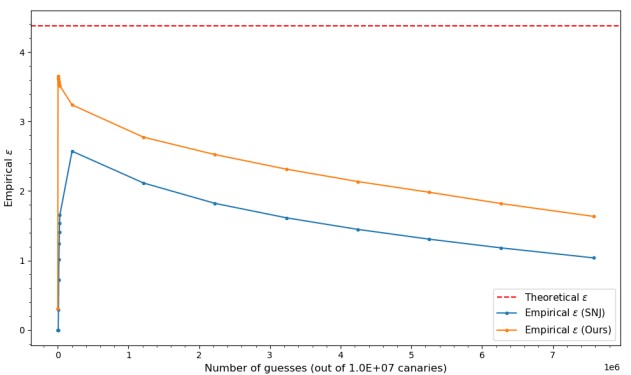

Figure 6. Effect of number of guesses (Gaussian mechanism with standard deviation 1.0). Empirical $\epsilon$ is reported at $\delta = 10^{-5}$.

Figure 7. Effect of number of guesses (Gaussian mechanism with standard deviation 2.0). Empirical $\epsilon$ is reported at $\delta = 10^{-5}$.

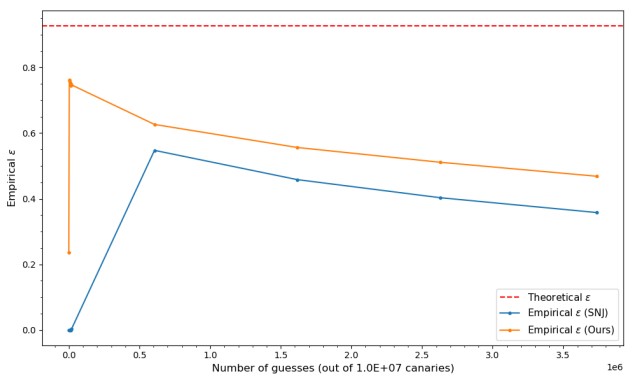

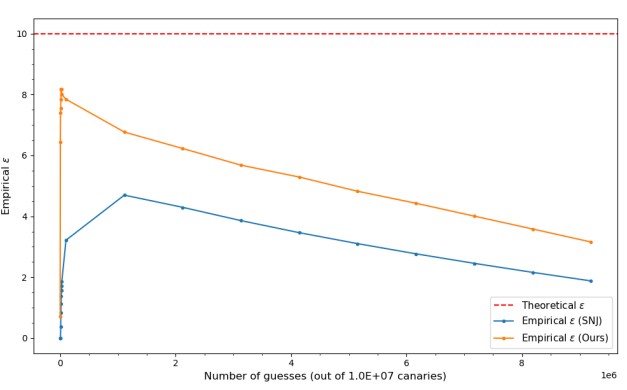

Figure 8. Effect of number of guesses (Gaussian mechanism with standard deviation 4.0). Empirical $\epsilon$ is reported at $\delta = 10^{-5}$.

Figure 9. Effect of number of guesses (Gaussian mechanism with standard deviation 0.5). Empirical $\epsilon$ is reported at $\delta = 10^{-5}$.

# D. Experimental details

**Idealized setting:** In the idealized setting, we work with a toy version of the mechanism to calculate the *expected* number of correct guesses for the ideal adversary. For Gaussian mechanism, the ideal setting for an adversary is when we have a Gaussian mechanism that is used to calculate the sum of vectors. In this setting, each canary represents a unit vector that is orthogonal to all other canary vectors. Then, given the noisy sum, the adversary will calculate the likelihood of the canary being used in the sum, and then decides on the guesses based on these likelihoods. For the setting that the adversary has more than 2 guesses $(k > 2)$, we use a slightly different idealized setting. In all settings, we run the attack 100 times and average the result to get the expected number of correct guesses. Algorithm 4 shows how we calculate the number of correct guesses in the idealized setting.

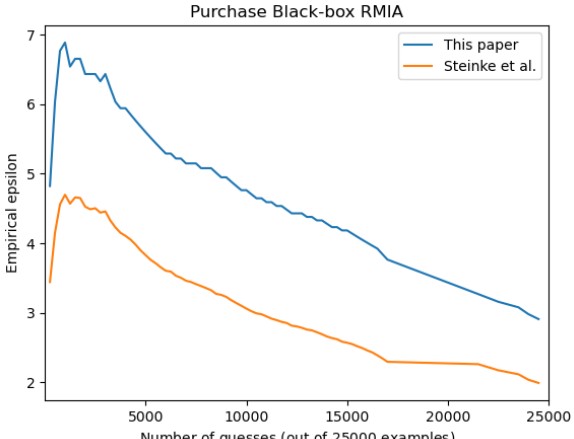

Figure 12. Comparison with auditing procedure of (Steinke et al., 2023) on non-private model trained on Purchase against black-box RMIA method (Zarifzadeh et al., 2023). Empirical $\epsilon$ is reported at $\delta = 10^{-5}$.

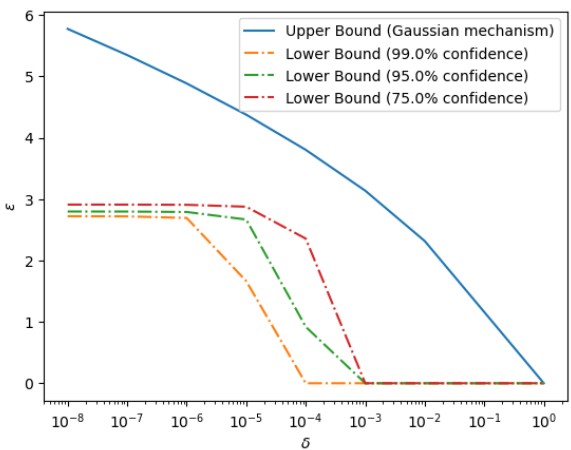

Figure 10. Idealized setting for different values of $\delta$ and confidence levels for bounds of (Steinke et al., 2023).

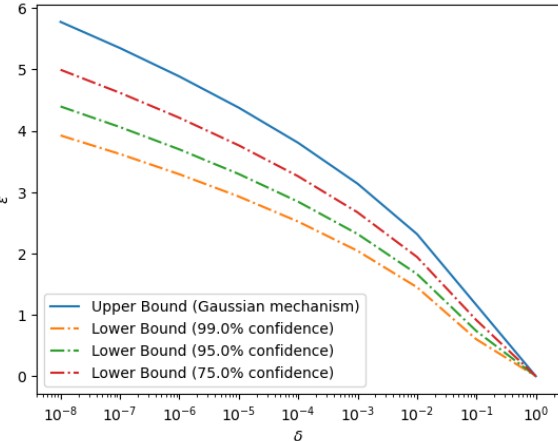

Figure 11. Idealized setting for different values of $\delta$ and confidence levels for our bounds.

**Algorithm 4** Simulate the Number of Correct Guesses

```python
import numpy as np
from scipy.special import softmax
from numpy.random import normal, binomial
def idealized_setting(target_noise, n_guesses, n_canaries, k):
    n_correct_vec = []
    if k==2:
        for _ in range(100):
            s_vector = binomial(1, 0.5, size=n_canaries) * 2 - 1
            noise = normal(0, 2*target_noise, n_canaries)

            noisy_s = s_vector + noise

            sorted_noisy_s = np.sort(noisy_s)

            threshold_c = sorted_noisy_s[-int(n_guesses)//2-1]
            n_correct = np.ceil(n_guesses*(s_vector[noisy_s > threshold_c] ==
            ↪  1).mean())

            n_correct_vec.append(n_correct)
    else:
        for _ in range(100):
            s_recon_vec = np.random.randint(0, k, n_canaries)

            s_vec_recn_ohe = np.eye(k)[s_recon_vec]
            s_recon_noisy_vec_ohe = s_vec_recn_ohe + normal(0,
            ↪  np.sqrt(2)*target_noise, s_vec_recn_ohe.shape)

            idx_max = np.argmax(s_recon_noisy_vec_ohe, axis=1)

            buckets = softmax(s_recon_noisy_vec_ohe/(2*target_noise**2),
            ↪  axis=1)[np.arange(s_recon_noisy_vec_ohe.shape[0]), idx_max]
            sorted_buckets = np.sort(buckets)
            bucket_c_thr = sorted_buckets[-int(n_guesses)]

            n_correct_rec = np.ceil(
                n_guesses*(s_recon_vec[buckets > bucket_c_thr] ==
                ↪  s_recon_noisy_vec_ohe[buckets >
                ↪  bucket_c_thr].argmax(1)).mean()
            )
            n_correct_vec.append(n_correct_rec)

    return int(np.array(n_correct_vec).mean(0))
```

## Auditing code

Here we include the code to compute empirical epsilon.

```python
from scipy.stats import norm
import numpy as np

# Calculate h and r recursively (no abstentions)
def rh(inverse_blow_up_function, alpha, beta, j, m, k=2):
    # Initialize lists to store h and r values
    h = [0 for _ in range(j + 1)]
    r = [0 for _ in range(j + 1)]
    # Set initial values for h and r
    h[j] = beta
    r[j] = alpha
    # Iterate from j-1 to 0
    for i in range(j - 1, -1, -1):
        # Calculate h[i] using the maximum of h[i+1] and a scaled inverse
        ↪  blow-up function
        h[i] = max(h[i + 1], (k - 1) * inverse_blow_up_function(r[i + 1]))
        # Update r[i] based on the difference between h[i] and h[i+1]
        r[i] = r[i + 1] + (i / (m - i)) * (h[i] - h[i + 1])
    # Return the lists of h and r values
    return (r, h)

# Audit function without abstention
def audit_rh(inverse_blow_up_function, m, c, threshold=0.05, k=2):
    # Calculate alpha and beta values
    alpha = threshold * c / m
    beta = threshold * (m - c) / m
    # Call the rh function to get the lists of h and r values
    r, h = rh(inverse_blow_up_function, alpha, beta, c, m, k)
    # Check if the differential privacy condition is satisfied
    if r[0] + h[0] > 1.0:
        return False
    else:
        return True

# Calculate h and r recursively (with abstentions)
def rh_with_cap(inverse_blow_up_function, alpha, beta, j, m,c_cap, k=2):
    h=[0 for i in range(j+1)]
    r=[0 for i in range(j+1)]
    h[j]= beta
    r[j]= alpha
    for i in range(j-1,-1,-1):
        h[i]=max(h[i+1],(k-1)*inverse_blow_up_function(r[i+1]))
        r[i]= r[i+1] + (i/(c_cap-i))*(h[i] - h[i+1])

    return (r,h)

# Audit function with abstentions
def audit_rh_with_cap(inverse_blow_up_function, m, c,c_cap, threshold=0.05,
↪  k=2):
    threshold=threshold*c_cap/m
```

```python
        alpha=(threshold*c/c_cap)
        beta=threshold*(c_cap-c)/c_cap
        r,h=rh_with_cap(inverse_blow_up_function, alpha, beta, c, m, c_cap, k)

        if r[0]+h[0]>c_cap/m:
            return False
        else:
            return True

# Calculate the blow-up function for Gaussian noise
def gaussianDP_blow_up_function(noise):
    def blow_up_function(x):
        # Calculate the threshold value
        threshold = norm.ppf(x)
        # Calculate the blown-up threshold value
        blown_up_threshold = threshold + 1 / noise
        # Return the CDF of the blown-up threshold value
        return norm.cdf(blown_up_threshold)
    return blow_up_function

# Calculate the inverse blow-up function for Gaussian noise
def gaussianDP_blow_up_inverse(noise):
    def blow_up_inverse_function(x):
        # Calculate the threshold value
        threshold = norm.ppf(x)
        # Calculate the blown-up threshold value
        blown_up_threshold = threshold - 1 / noise
        # Return the CDF of the blown-up threshold value
        return norm.cdf(blown_up_threshold)
    return blow_up_inverse_function

# Define a function to calculate delta for Gaussian noise
def calculate_delta_gaussian(noise, epsilon):
    # Calculate delta using the formula
    delta = norm.cdf(-epsilon * noise + 1 / (2 * noise)) - np.exp(epsilon) *
    ↪  norm.cdf(-epsilon * noise - 1 / (2 * noise))
    return delta

# Define a function to calculate epsilon for Gaussian noise
def calculate_epsilon_gaussian(noise, delta):
    # Set initial bounds for epsilon
    epsilon_upper = 100
    epsilon_lower = 0
    # Perform binary search to find epsilon
    while epsilon_upper - epsilon_lower > 0.001:
        epsilon_middle = (epsilon_upper + epsilon_lower) / 2
        if calculate_delta_gaussian(noise, epsilon_middle) > delta:
            epsilon_lower = epsilon_middle
        else:
            epsilon_upper = epsilon_middle
    # Return the upper bound of epsilon
    return epsilon_upper

# Get the empirical epsilon value
```

```python
def get_gaussian_emp_eps_ours(candidate_noises, inverse_blow_up_functions, m,
↪  c, threshold, delta, k=2):
    # Initialize the empirical privacy index
    empirical_privacy_index = 0
    # Iterate through candidate noises until the privacy condition fails
    while audit_rh(inverse_blow_up_functions[empirical_privacy_index], m, c,
    ↪  threshold=0.05, k=k):
        empirical_privacy_index += 1
    # Get the empirical noise and calculate the empirical epsilon
    empirical_noise = candidate_noises[empirical_privacy_index]
    empirical_eps = calculate_epsilon_gaussian(empirical_noise, delta=delta)
    # Return the empirical epsilon
    return empirical_eps

# Set target noise and generate candidate noises
target_noise = 0.6

candidate_noises=[target_noise+ i*0.01 for i in range(1000)]
inverse_blow_up_functions=[gaussianDP_blow_up_inverse(noise) for noise in
↪  candidate_noises]
```

