# OpenReview forum: "Auditing $f$-differential privacy in one run"
_ICML.cc/2025/Conference — ICML 2025 oral_

### Official Review · Reviewer_vgFz · 2025-03-11

**Overall Recommendation:** 4

**Summary:**

This paper considering auditing the claimed $f$-DP guarantees of a model training procedure with a single training run; prior work of Steinke et al. (NeurIPS 2023) considering the same problem but in the setting of $(\epsilon, \delta)$-DP. $f$-DP provides a more fine-grained view of the DP guarantees of a mechanism, instead of a crude understanding with two parameters $(\epsilon, \delta)$.

Given a mechanism $M$ that is claimed to satisfy a certain $f$-DP guarantee, the auditing procedure is formalized as a two-step procedure:
* The auditor designs a randomized experiment using the description of $M$ that yields some observation $o$; this involves running $M$ on some designed input dataset.
* The auditor then based on the observation $o$ and claimed $f$-DP, either accepts or rejects.

The guarantee that such an auditor must satisfy is that if $M$ indeed satisfies $f$-DP, then the auditor accepts with high probability.
The utility of the auditor is established empirically, namely if $M$ does not satisfy $f$-DP (or violates it quite significantly), then we would like to reject with high probability.

Such an auditor can be used to compute an empirical privacy guarantee by considering a family $F$ of potential $f$-DP guarantees, and given the observation, picking a maximal $f$ which the auditor accepts, and report the $(\epsilon, \delta)$-DP guarantee satisfied by it.

The procedure considered by Steinke et al. was to consider $m$ canaries, and insert each in the training dataset with probability $0.5$, and the auditor then uses a membership inference technique to guess if each canary was included in the training set or not.

The main contributions in the paper are:
* It designs a variant of the auditing procedure of Steinke et al. that uses $k$ choices for each of $m$ canaries inserted, instead of just a single choice which is included or not, and
* It provides an upper bound on the expected number of correct guesses in terms of the $f$-DP guarantee and designs a hypothesis test using this bound.
* Experimental results are provided, on simple Gaussian mechanism, and on CIFAR dataset, that demonstrate that the proposed auditing procedure is able to provide a large empirical privacy lower bound.

The main advantage of using $f$-DP instead of $(\epsilon, \delta)$-DP is that in Steinke et al. handled the $\delta$ term naively by making the bound worse by $m \delta$, which makes the bound degrade with the number of canaries added. Using $f$-DP is able to handle this in a more fine-grained manner.

**Claims And Evidence:**

The claims are supported by sufficient evidence in the paper.

**Essential References Not Discussed:**

To the best of my knowledge, essential references are adequately discussed.

**Experimental Designs Or Analyses:**

The experimental design makes sense to me and I do not see any issues.

**Methods And Evaluation Criteria:**

The proposed methods are sound, and the evaluation criteria also make sense for the considered problem.

**Other Comments Or Suggestions:**

* Update `\icmltitlerunning` command to have the title of the paper instead of “Submission and Formatting Instructions for ICML 2025”.
* Line 132 (left column): What is $X$ in $f : X \to \mathbb{R}$ ? Shouldn’t it be $f : [0, 1] \to [0, 1]$ ?
* Line 188 (left column): Is there a $\log$ missing in the equation for $\epsilon(\delta)$?
* Perhaps move Algorithm 1 to Page 4 where it is first mentioned?
* I found Algorithm 3 extremely hard to understand. Perhaps it is trying to perform the test in some efficient manner, but if possible, I would recommend preferring simplicity over efficiency. Is there a different procedure that could be inefficient, but is more intuitive to understand? If so, I would recommend including such a version in the main body, and defer this more efficient version to the Appendix. Or at the very least, it would be helpful to explain what $r[i]$ and $h[i]$ are supposed to intuitively stand for.
* Figure 1: What is the value of $\delta$? Is it the same value used for all the four noise values?
* Figure 4: It might be better to stick to the convention of using `blue` for Steinke et al. and `orange` for “This paper” to be consistent with Figures 1-3.

**Other Strengths And Weaknesses:**

The paper makes a solid contribution for auditing mechanisms that satisfy $f$-DP. It is also well-written for the most part and a pleasure to read (modulo some some parts that I found hard to read that are mentioned under "Other Comments Or Suggestions" below).

I don't see any major weakness, but I feel there is one argument missing about handling a large family $F$ of potential $f$-DP guarantees. (See under "Questions for Authors".)

**Questions For Authors:**

One major confusion I have is: why it is correct to simply take the maximal $f \in F$.
Why is there no price to pay for how large the family $F$ is?

Naively, one could apply a union bound over all $F$ to argue that one does not accidentally pick an $f$ that the mechanism does not satisfy. But perhaps the union bound is not required if $F$ is totally ordered, since we are using the same observation $o$ in $\mathrm{evaluate}(o, f)$ for all $f \in F$, and perhaps it holds that for any $o$, if $f_1 \le f_2$ (i.e. $f_2$ is more private than $f_1$) then $\mathrm{evaluate}(o, f_1) \ge \mathrm{evaluate}(o, f_2)$ (this is a natural assumption on $\mathrm{evaluate}$, and perhaps better to state explicitly).

But when $F$ is not totally ordered, how does one argue this? I see that a full union bound over $F$ is not required, but some smarter union bound is perhaps needed. As far I can tell, the main body of the paper only analyzes the case of auditing a single $f$.

**Relation To Broader Scientific Literature:**

The paper makes a significant contribution to the literature on differential privacy, and is likely to inspire follow-up work.

**Theoretical Claims:**

Proofs of theoretical claims are provided in the Appendix (unfortunately, I was not able to go through all of it in detail, but I can believe they are correct).

---

### Official Review · Reviewer_xS1M · 2025-03-11

**Overall Recommendation:** 5

**Summary:**

This paper presents an accurate and efficient auditing procedure to assess the privacy level of mechanisms within the framework of $f$-DP. The authors utilize a more generalized and refined privacy notion, $f$-DP, and effectively solve the guessing game, a generalized framework for reconstruction and membership inference attacks. The theoretical derivations are sound and well-presented, and the experimental evaluations are well-designed and convincing.

**Claims And Evidence:**

Yes

**Essential References Not Discussed:**

N/A

**Experimental Designs Or Analyses:**

Given that this is a theoretical paper, a set of not so extensive experiments in "idealized" setting is convincing enough to me to illustrate the performance.

**Methods And Evaluation Criteria:**

The proposed methods are well-motivated and empirically validated.

**Other Comments Or Suggestions:**

This is a technically rich paper, and adding more intuition and explanations in the main body would enhance clarity. For example,
- More explanation of Definition 2.2 and Proposition 2.3 would be helpful. Initially, the connection to standard $(\epsilon,\delta)$-DP is clear, but the link to $f$-DP is less obvious. Perhaps explicitly stating that $\hat{f}$ is the conjugate $f^*$ (as in Dong et al.) and that the privacy curve of an $f$-DP mechanism is given by $\delta(\epsilon) = 1 +f^*(-e^{\epsilon}) = 1 + \sup_x -e^{\epsilon}x -f(x)$  would make this clearer.
- More clarification on Definition 2.7 would be helpful.

**Other Strengths And Weaknesses:**

Strengths:
1. The proposed method is efficient, accurate, and theoretically elegant.
2. To the best of my knowledge, this is the first work on one-shot $f$-DP auditing.
3. The method generalizes/refines results in the literature and demonstrates strong empirical performance.

Weaknesses: while Theorem 3.1 is an iterative-type result, which I don't quite like usually, I find its proof very cute. The utilization of this structure to derive Theorem 3.2 is particularly appealing. While a more direct bound would be interesting, the current result is already strong.

**Questions For Authors:**

1. regarding Def. 2.7, I assume the max over $x$ relates to the privacy curve/$f$-DP definition, while taking the min over $\epsilon$ provides the best possible valid tradeoff function. Is that right?
2. I also wonder (a). why use $\epsilon(\delta)$ instead of $\delta(\epsilon)$? (b) Is $\epsilon(\delta)$ essentially the best possible $e^{\epsilon}$ for a given $\delta$ in this def.?
3. Is Theorem 3.1 general for both guessing games? I assume it is general, but the context could be clearer.

Two questions out of curiosity:
4. Can we potentially leverage the structural property of attach methods to derive tighter bounds, i.e., attack-specific guessing game bounds instead of general bounds?
5. Suppose we call $K$ times instead of once—how would this impact the bounds? It seems that modifying Lemma A.1 would suffice, but I don't have a good intuition on what it would look like.

**Relation To Broader Scientific Literature:**

This is a novel contribution, providing an accurate and efficient auditing procedure for privacy assessment within the $f$-DP framework. It also generalizes and refines results in the literature. Also, to my best of knowledge, it is the first work on one shot $f$-DP auditing,

**Theoretical Claims:**

Yes. Please refer to Question and Comments sections below.

---

### Official Review · Reviewer_BXCR · 2025-03-13

**Overall Recommendation:** 3

**Summary:**

This paper looks at auditing f-DP in one run (adding to prior works like auditing DP in one run by Steinke et al) as opposed to one $\epsilon,\delta$-pair, providing tighter privacy leakage bounds and characterizes the privacy bounds of approximate DP mechanisms like the Gaussian mechanism better when the failure probability $\delta$ is non-zero. The authors do this via membership inference and a quasi-reconstruction based approach and provide theoretical results that attest to the soundness of their guarantees and discuss why they surpass their baselines in terms of closing the gap between empirically determined and theoretical values of $\epsilon$.

**Claims And Evidence:**

Yes they are, especially with empirical guarantees that attest to why this provides a better audit than the baseline (Steinke et al, cited in the paper) along with a detailed discussion on why they do better when $\delta>0$. The theoretical guarantees also provides grounds for soundness.

**Essential References Not Discussed:**

I think that the authors have done a wonderful job of listing all relevant references, including all baselines and state of the art attacks.

**Experimental Designs Or Analyses:**

I did. It is decent, the experiments include experiments on a standard dataset (CIFAR-10) and model (WideResNet16-4).

However, the setup of their results in Sec 4.1 (Subsubsection on Simple Gaussian Mechanism) is not defined well and is vague and needs to be defined.

In addition, for experimental rigor and robustness of their results, it would be advisable to have more results than for just one dataset and one model. Intuitively, I think the findings should generalize but for empirical rigor and to ensure it, I need to see those results.

**Methods And Evaluation Criteria:**

Yes they do. The attacks selected for the audit make sense and the datasets they test on are standard datasets used widely for such investigations.

**Other Comments Or Suggestions:**

* It will be helpful if the authors spend a bit more time discussing their theorem statements in a high-level manner and explain to the reader what each aspect of it does. Otherwise, it's harder to understand, especially for newcomers or not-as-much theoretically-inclined readers.

**Other Strengths And Weaknesses:**

* Strength: The authors anticipate all the questions they would need to answer while justifying their approach and take a principled and exhaustive approach to justifying their novelty, the merits of their method over the baselines, and their design choices/procedures.
* Weakness 1: The setup for the experiment on the Simple Gaussian Mechanism in Sec 4.1 is very vaguely described and it is not clear what is going on (Which model is being used? Is the data synthetic?). Even if this subsection uses the setup from another paper, which I assume it is, then the authors should describe it here (or at least in the appendix) nonetheless for self-containment and as a good practice.
* Weakness 2: The authors define an auditing method based on reconstruction (or an approximation of it, so to speak) but never use it in their experiments or provide results on it, as it appears.

**Questions For Authors:**

* Pardon me if I missed it somewhere, but where do you describe the candidates for $f$, $h_i$ rigorously? I know you mention them being curves for the Gaussian mechanism but some explicit discussion on what they look like would be very helpful. I did not find it based on my reading of the text, and that seems like an important detail to have.
* Can you please address Weakness 1?
* I know that this is a theoretical work but it would be really nice to have results on more than one model-dataset pair (I see results only on WideResNet and CIFAR-10). It is good practice to see that empirical results generalize across settings. I think the empirical results are the key convincing factor about the improvement over the baseline.

This is a strong work but there are some issues and I need these to be clarified. If you answer these questions satisfactorily, I *will* raise my score to an accept, if not more.

* Bonus question: Can you please explain why (as it seems) that you defined a reconstruction-based approach but never used it in your experiments? What differences in the quality of $\epsilon$ estimation will it yield over the MIA-based approach? If you do not intend to add experimental results on the reconstruction-based approach (correct me if I'm wrong), then I find the inclusion of the discussion about reconstruction-based auditing redundant; surely in that case simply talking about MIA-based auditing is sufficient and I wouldn't count the reconstruction-based auditing approach as a merit/full contribution. Please clarify.

**Relation To Broader Scientific Literature:**

This is a neat contribution and adds to literature on efficiently auditing models for DP guarantees, building upon the popular work by Steinke et al. This, by auditing f-DP instead of $(\epsilon,\delta)$-DP, provides a more general and tight privacy audit, and fills the gap in the literature for finding $f$ for $f$-DP based auditing for an ML model.

Steinke, Thomas et al. “Privacy Auditing with One (1) Training Run.” ArXiv abs/2305.08846 (2023): n. pag.

**Theoretical Claims:**

The authors do provide proofs for their theoretical claims, but due to their length and time constraints, I was not able to go through them. For the time being, I'll take their correctness in good faith but might revisit them later and ask any questions I may have.

---

### Official Review · Reviewer_cWiv · 2025-03-14

**Overall Recommendation:** 2

**Summary:**

The paper studies auditing of DP parameters $\varepsilon$ and $\delta$ using one training run. The goal is to find high-confidence lower bounds for the DP parameters via membership inference guessing game, similarly to the baseline method by [Steinke et al., 2023](https://proceedings.neurips.cc/paper_files/paper/2023/file/9a6f6e0d6781d1cb8689192408946d73-Paper-Conference.pdf). In this guessing game randomly selected "auditing samples" are included from a given "auditing set" to the actual training set, and then the goal is to try to correctly guess which ones of the auditing samples were included and which ones were not. Based on the number of correc guesses, lower bounds for the DP parameters can be obtained. In the baseline method by [Steinke et al., 2023](https://proceedings.neurips.cc/paper_files/paper/2023/file/9a6f6e0d6781d1cb8689192408946d73-Paper-Conference.pdf), point-wise ($\varepsilon,\delta$)-lower bounds are given. Here the novelty is to obtain $f$-DP lower bounds, i.e., to determine a $f$-DP curve which would be an upper bounding trade-off curve.

**Claims And Evidence:**

Yes, the experiments clearly show that the $f$-DP based approach leads to higher ($\varepsilon,\delta$)-DP lower bounds than point-wise estimates by [Steinke et al., 2023](https://proceedings.neurips.cc/paper_files/paper/2023/file/9a6f6e0d6781d1cb8689192408946d73-Paper-Conference.pdf).

**Essential References Not Discussed:**

I cannot think of such references.

**Experimental Designs Or Analyses:**

I read the description of the experiments, I think everything makes sense. I could not find what is the value of $\delta$ used for Figure 1 to 4, but I assumed it is $10^{-5}$ as that is used to illustrate the theoretical upper bounds of the experiments behind Fig. 2 to 4.

**Methods And Evaluation Criteria:**

Yes, I think this selection of experiments well illustrate the benefits of this approach.

**Other Comments Or Suggestions:**

Consider a quick numerical experiment for directly combining GDP auditing and the method of Steinke et al. using the code given in their [Appendix](https://arxiv.org/pdf/2305.08846) :

First, adjust the number of correct guesses so that $\varepsilon$-values are similar as given in your Figure 2 for the baseline method. Then, using their code, compute the $\varepsilon$-lower bounds using different values of $\delta$ using the same $p$-value 0.05, and find the smallest $\mu$-GDP parameter for which the GDP $\delta(\varepsilon)$-curve is above those lower bounds (the touch point is somewhere near $\delta=10^{-3}$). Then, compute the $\varepsilon$-lower bounds for $\delta=10^{-5}$ using the GDP curve. The resulting difference to the original point-wise estimate seems to be very close to the difference that shows up in your Figure 2. This makes me think that the improvement comes only from using $f$-DP auditing, and that the analysis here is somehow very similar to that of Steinke et al. I.e., by combining their point-wise method and the $f$-DP accounting by Nasr et al. (2023), similar differences seem to be obtainable.

If it is the case, and there are such equivalences, I think that should be clearly written out and analyzed, and the actual contributions of the paper should be stated more clearly.

Somehow my overall impression is that here is some interesting contribution, but it remains unclear what it exactly is as the paper is not yet that mature.

**Other Strengths And Weaknesses:**

- The general idea of combining the guessing game approach by Steinke et al. and $f$-DP accounting proposed by Nasr et al. (2023) is nice and clearly improves upon the baseline method by Steinke et al.

Weaknesses:

- One has to of course keep in mind that the improvement relies on the assumption that the underlying method actually has a trade-off curve that belongs to the family of trade-off curves that are used here for the auditing, that is the additional contribution here. E.g., if carry out GDP auditing, i.e., the family of trade-off curves is the trade-off curves of GDP mechanisms, then it is possible that one ends up to wrong conclusions about DP lower bounds in case the underlying mechanisms trade-off curve is far from a GDP trade-off curve.

- The presentation could be improved a lot. It is very difficult to follow the text sometimes and there are also typos here and there. A lot is left unsaid, e.g., about GDP auditing. I could see the GDP auditing formulas gives only in the code of the appendix.

- Due to the not so clear presentation, the connection to the baseline method by Steinke et al. remains a bit unclear. My impression is that this paper is kind of repeating the analysis by Steinke et al. using the $f$-DP formalism, so that we have $f$-DP lower bound directly at hand in the end. However, I believe one can get to similar conclusions by using the point-wise lower bounds given by the method of Steinke et al., then given a family of $f$-DP curves, find lowest $f$-DP curve that is below the trade-off functions determined by these point-wise estimates. When thinking about ($\varepsilon,\delta)$-bounds, e.g. in case of GDP auditing, one just finds the smallest $\mu$-GDP parameter $\mu$ for which the privacy $\delta(\varepsilon)$-curve is above all the point-wise lower bounds.

- After all, this is also average case analysis like the auditing method by Steinke et al. I believe this corresponds to threshold membership auditing with the median score value or something close to that.

**Questions For Authors:**

Considering the above small experiment indicating that it is possible to get similar $\varepsilon$-lower bounds directly by combinign the method of Steinke et al. and GDP auditing, what is exactly the novelty here and what are the most important contributions? Somehow your $f$-DP based analysis looks much cleaner. However, in addition to possibly simplifying the analysis of Steinke et al., is there some clear benefit in using this approach (i.e., your Theorem 3.1 and Algorithm 3) ? You also mention there are improvements in the reconstruction attack result, compared to the result of Hayes et al (2023). But does that remain just as a theoretical contribution here?

**Relation To Broader Scientific Literature:**

This paper is very closely related to the paper by [Steinke et al., 2023](https://proceedings.neurips.cc/paper_files/paper/2023/file/9a6f6e0d6781d1cb8689192408946d73-Paper-Conference.pdf). Even up to the point that I would think that there is not that great novelty compared to that paper. In a sense, this paper is taking the $f$-DP auditing proposed by [Nasr et al., 2023](https://www.usenix.org/system/files/usenixsecurity23-nasr.pdf) and combining it with the point-wise lower bounding technique by [Steinke et al., 2023](https://proceedings.neurips.cc/paper_files/paper/2023/file/9a6f6e0d6781d1cb8689192408946d73-Paper-Conference.pdf).

**Theoretical Claims:**

I did not read in details all the proofs in the appendix, but I read the main text and I was able to follow it and I think everything makes sense. I have no reason to believe there are errors that would change the conclusion.

---

### Decision · Program_Chairs · 2025-05-01

**Decision:**

Accept (oral)

**Comment:**

The reviewers collectively recognized this paper's contribution in developing a more accurate and efficient auditing procedure for f-differential privacy that requires only a single run of the target mechanism. Reviewers highlighted the theoretical elegance of the paper, particularly its generalization of previous work and improved handling of the delta term when auditing privacy guarantees. For the camera-ready version, please clarify the experimental setup details, especially regarding the Gaussian mechanism experiments, and ensure consistent notation throughout the paper.